# How to Avoid Debate:
# Scalable AI Safety via Doubly-Efficient Interactive Proofs

Liyan Chen [1]   Yael Tauman Kalai [1]   Zoe Xi [1]

## Abstract

As AI models continue to develop powerful capabilities, it becomes critical that we are able to verify that their output is aligned with our intentions. A recent line of work focuses on verification via debate, a model of interactive proofs where two competing powerful provers, or AI models, debate each other to convince a weak verifier, or a human, of the correctness of their claim. However, debate assumes that the two AI models possess equal abilities and that one of them is truthful, which may not be realistic.

In this work, we show *how to avoid debate*: we initiate the study of *single-prover* interactive proofs for AI safety. Prior results in single-prover interactive proofs do not immediately carry over to the AI safety setting because they do not work when the computation has access to an oracle, such as to human judgment or an external database such as the web. We present doubly-efficient single-prover interactive proofs for oracle-aided computations (also known as relativizing proofs), in the settings where (1) the computation is robust, in the sense that the output does not change if at most a small fraction of the answers to oracle queries are incorrect, or (2) the oracle is a low-degree polynomial. These results suggest that interactive verification is possible even without debate, under structured or noise-tolerant oracle access.

## 1. Introduction

As machine learning models become increasingly capable, they are being trained to perform complex tasks that would be prohibitively expensive for a human to verify, leading to critical safety concerns. How can we efficiently supervise the training of an AI model to perform some complex task in a way that aligns with our intentions, when we may only have an obscure understanding of the task ourselves?

As a concrete example, consider the scenario of training a large language model (LLM) to write a long high-stakes document, such as a legal contract.[1] To generate a training label on a contract produced by the LLM, it is necessary to verify that every passage of the contract is correct, where correctness here is dictated by human judgment. However, it would be unreasonably expensive to require a human to carefully read through a long legal contract to produce just one training label. In this setting, it is important to have a training protocol that is extremely efficient in its use of human judgments.

Various approaches to this problem, known as *scalable oversight*, have been proposed in the AI safety literature (Christiano et al., 2018; Leike et al., 2018; Irving et al., 2018). In particular, a recent line of work focuses on designing verifiable training protocols by leveraging tools from theoretical cryptography, namely the area of interactive proofs (Irving et al., 2018; Brown-Cohen et al., 2024; 2025). An interactive proof (Goldwasser et al., 1985; Babai, 1985) is a protocol wherein a powerful but untrusted prover interacts with a weak verifier to try to convince the verifier of the validity of some statement. For security, if the statement is indeed valid, then the verifier should accept, and if the statement is not valid, then the verifier should reject (with high probability). At a high level, taking the prover to be an AI model and the verifier to be a human, this closely resembles what we want in our safe training setting.

The main reason why prior work in interactive proofs does not immediately carry over to our setting, however, is that these classical results assume that the computation is given in a well-defined mathematical manner, such as a Turing machine. In the setting of AI safety, tasks can be modeled as computation with access to an oracle, which is a "black-box" function that we can't see the inner workings of; an oracle might represent human judgment or an external database such as the web, for example. Going back to our legal

---

[1]Massachusetts Institute of Technology. Correspondence to: Zoe Xi <zoexi@mit.edu>.

*Proceedings of the $43^{rd}$ International Conference on Machine Learning*, Seoul, South Korea. PMLR 306, 2026. Copyright 2026 by the author(s).

---

[1]This example is due to Brown-Cohen, Irving, and Piliouras (Brown-Cohen et al., 2024).

document example, we would like to have a protocol where the AI model (the prover) makes few queries to a human expert, which we could model as an oracle. Interactive proofs for oracle-aided computations are said to *relativize*, and we do not have interactive proofs that relativize. At the same time, relativization is believed to be essential to AI safety: the UK AISI Alignment team claims that "in order to be relevant to AI safety, an interactive proof result must relativize" (Irving & Marshall, 2025).

This aim of relativization has driven researchers to study a new model of interactive proofs, known as *debate*. In this model, introduced by Irving, Christiano, and Amodei in their 2018 paper "AI Safety via Debate" (Irving et al., 2018), there are two debating provers, who hold opposing claims, and a verifier, and the provers debate each other to try to convince the verifier of the validity of their claim. In the same paper, Irving, Christiano, and Amodei show that DEBATE = PSPACE (where DEBATE is the class of problems that have a debate interactive proof), and this result relativizes, whereas the celebrated IP = PSPACE result (Shamir, 1992) (where IP is the class of problems that have a single-prover interactive proof), does not; the moral is that to recover relativization one can introduce a second debating prover. A follow-up work by Brown-Cohen, Irving, and Piliouras (Brown-Cohen et al., 2024) introduces an analogue of this model of debate, called *doubly-efficient* debate, where the provers are constrained to run in polynomial time instead of being computationally unbounded—the motivation is that in our AI safety setting the provers are AI models, and AI models, while powerful, are still computationally bounded.

However, debate relies on assumptions that may not be realistic: in order to use debate for AI safety, we need that the two AI models are properly incentivized to debate, that they have the same computational abilities, and that one of them is truthful. This last assumption of truthfulness is particularly concerning: it hinges on the assumption that it is harder to lie than to refute a lie, that is, that one of the AI models is incentivized to be truthful, which may or may not be true in any particular setting (Irving et al., 2018). Further limitations of debate as an approach for AI alignment are discussed in (Irving et al., 2018).

In light of these limitations, in this work we investigate the following question:

*Can we construct a single-prover interactive proof that relativizes?*

In the model that we study, there is a polynomial-time prover and an even more efficient verifier, who both have access to an oracle, and the prover tries to convince the verifier of the correctness of some computation that may depend on queries to the oracle. Notably, the super-efficient verifier not only runs in much less time than the prover, but he also makes few queries to the oracle, much fewer than the number that he would need in order to perform the computation by himself—in the legal contract example, this would mean that very few queries to the human expert are needed.

Unfortunately, Barbara, Chiesa, and Guan (Barbara et al., 2025) prove that there do not exist interactive proofs that relativize for all computations. Intuitively, their observation is that if the computation uses many oracle gates to a random oracle, and the output of the computation is unstable, in the sense that it hinges on the answers of all the oracle queries being correct, then the verifier would need to check the correctness of every oracle call, and thus would not be efficient.

### 1.1. Our results

In this paper, we construct relativizing doubly-efficient interactive proofs for two natural settings, where the computation is "robust" or where the oracle is a low-degree polynomial. Our results suggest we can recover relativization even without two debating provers.

**Robust computation.** We say that a computation is robust if the output does not change even if a small fraction of answers to oracle queries are modified. Intuitively, robustness allows a verifier to "spot-check" oracle calls instead of having to evaluate all of them. It seems natural to assume that the computation is robust, especially in the setting where the oracle is taken to represent human judgment: human judgment is already error-prone, so as a safeguard, the output of the computation should not change if a small number of the human answers change, otherwise the output is not very meaningful. More broadly, many natural tasks may be able to be made robust using redundancy, e.g., in the case of human judgment, we might rephrase the question or ask multiple people and take the majority answer.

**Low-degree oracles.** We also consider computations with access to an oracle that can be represented by a low-degree polynomial. Unlike the above robust setting, here we do not assume anything about the computation, but rely on the strong algebraic structure of the oracle in order to make the verifier efficient. Our protocol in this setting can be viewed as a first step towards designing protocols for computations with access to an oracle that is "learnable" by a simple machine learning model class. More concretely, we might take the oracle to represent a database, such as the web: any database $\{0,1\}^{\log N} \to \{0,1\}$ can be converted into a low-degree oracle $\mathbb{F}^{\log N} \to \mathbb{F}$ via a standard low-degree extension (see Chapter 3 of (Thaler, 2022)). Then our protocol would allow us to efficiently verify the correctness of any computation that can interact with this database.

Our protocols exhibit different tradeoffs between soundness

and efficiency guarantees. Some protocols achieve statistical soundness, which guarantees security against even computationally unbounded cheating provers, and some achieve only computational soundness, which guarantees security against polynomial-time cheating provers. The protocols have varying efficiency guarantees regarding the number of bits communicated between the prover and the verifier, the running time of the verifier, and the number of queries that the verifier makes to the oracle. For a more detailed explanation of these notions, see Section 2. Also see Table 1 for an overview of our results.

**Outline of the paper.** We begin in Section 2 by introducing our model of doubly-efficient interactive proofs for oracle-aided computation and more definitions that we need. In Section 4 and Section 5, we present our results for the settings where the computation is robust and the oracle is a low-degree polynomial, respectively. We end in Section 6 with some open questions.

### 1.2. Related work

**AI safety via debate.** The work most closely related to ours is Brown-Cohen, Irving, and Piliouras' "Scalable AI Safety Via Doubly Efficient Debate," which presents doubly-efficient *debate* protocols for oracle-aided computation (Brown-Cohen et al., 2024); we study doubly-efficient *single-prover* protocols for oracle-aided computation. Debate was first proposed as an approach to scalable oversight in 2018 by Irving, Christiano, and Amodei in their paper "AI Safety via Debate" (Irving et al., 2018), which assumes that the two debating provers are computationally unbounded. More recently, work on debate has focused on limitations, e.g., the "obfuscated arguments problem," wherein a dishonest debater can come up with a flawed argument where the flaw is very hard to find (Barnes, 2020; Brown-Cohen et al., 2025; Buhl et al., 2025).

**Delegating computation.** The problem of supervising complex computation with a weak verifier has been studied extensively in computational complexity and cryptography. Besides the celebrated IP = PSPACE theorem (Shamir, 1992) and the PCP theorem (Arora & Safra, 1998), there has been a line of work studying doubly-efficient interactive proofs, motivated by the application of delegating computation (Goldwasser et al., 2015; Reingold et al., 2016). Currently, Berger, Goyal, Hong, and Kalai (Berger et al., 2025) give the best-known result for delegating polynomial-space computation. On the other hand, if we only consider security against polynomial-time algorithms, following the seminal work of Kilian and Micali (Kilian, 1992; Micali, 2000), there has been a long line of works studying super-efficient verification for any NP statement (Valiant, 2008; Bitansky et al., 2013; Kalai et al., 2014; Ben-Sasson et al.,

2018; Gabizon et al., 2019; Kothapalli et al., 2022; Setty et al., 2024; Jin et al., 2024; 2025), known as SNARGs or SNARKs, which are already widely used in practice (see, e.g., (Ben-Sasson et al., 2014)). Our work can be regarded as an extension of both of the above lines, to the setting of oracle-aided computation.

## 2. Preliminaries

We let $\mathbb{N} = \{1, 2, 3, \ldots\}$ denote the set of natural numbers, and for $n \in \mathbb{N}$, we write $[n]$ for the set $\{1, \ldots, n\}$. We use $\{0, 1\}^n$ for the set of binary strings of length $n$ and $\{0, 1\}^*$ for the set of binary strings of any length.

**(Oracle) Turing machines.** In our protocols, we model the prover and verifier as oracle Turing machines. On input a binary string $x$, a ***Turing machine*** $M : \{0, 1\}^* \to \{0, 1\}$ outputs a yes-or-no answer encoded as a bit $M(x)$. An ***oracle Turing machine*** $M$ with access to ***oracle*** $O : \{0, 1\}^* \to \{0, 1\}$, which we denote by $M^O$, additionally has the ability to write a query to $O$ and get back an answer in a single step.

**(Oracle) Boolean circuits.** We model the computation that the prover and verifier perform as an oracle Boolean circuit, which is a diagram that shows how to obtain an output bit from a binary input string by applying some sequence of OR ($\vee$), AND ($\wedge$), and NOT ($\neg$) operations and calls to the oracle. Here is a more formal definition:

*Definition* 2.1 ((Oracle) Boolean circuit). For every $n \in \mathbb{N}$, an $n$-input, single-output Boolean circuit is a directed acyclic graph with $n$ *sources*, i.e., vertices with no incoming edges, and one *sink*, i.e., a vertex with no outgoing edges. Every non-source vertex is called a *gate* and is labelled with one of $\vee$, $\wedge$, or $\neg$; $\vee$ and $\wedge$ gates have two incoming edges and $\neg$ gates have one incoming edge. For some input string $x \in \{0, 1\}^n$, the output of the $i$th source vertex is the $i$th bit of $x$, and the value of a gate is defined recursively as the result of applying the logical operation of the gate on the values of its *children*, i.e., the vertices with an edge going into the gate. The *output* of the circuit on $x$ is the output of the sink vertex.

The *size* of a circuit is the number of gates it contains, and the *depth* is the length of the longest path from a source vertex to the sink vertex.

An oracle Boolean circuit with respect to oracle $O : \{0, 1\}^* \to \{0, 1\}$ additionally has oracle gates, which have an arbitrary number of incoming edges. We view the input to an oracle gate as a string given by the values of its children. The value of an oracle gate is the value of the oracle applied to its input string.

We say that an oracle circuit makes ***adaptive*** oracle queries

*Table 1.* An overview of our results. We model the computation as a layered oracle Boolean circuit containing $d$ layers of $\ell$ oracle gates. Here $\mathcal{V}_{\text{time}}$ is the verifier's running time, $\mathcal{V}_q$ is the number of oracle queries made by the verifier, and $\varepsilon$-robust means that the output of the circuit does not change even if at most an $\varepsilon$-fraction of the answers to oracle queries are changed on any input (see Section 2 for more formal definitions).

| Protocols | Soundness | $V_{\text{time}}$ | $V_q$ | Setting | Adaptive? |
|---|---|---|---|---|---|
| Theorem C.1 | statistical | $O(\varepsilon\ell n + 1/\varepsilon)$ | $O(1/\varepsilon)$ | $\varepsilon$-robust | no |
| Theorem C.2 | statistical | $\text{poly}(d, D, \log S)(\varepsilon S + 1/\varepsilon)$ | $\text{poly}(d)/\varepsilon$ | $\varepsilon$-robust | yes |
| Theorem C.3 | computational | $O(1/\varepsilon + n)$ | $O(1/\varepsilon)$ | $\varepsilon$-robust | yes |
| Theorem D.1 | computational | $O(n)$ | 1 | poly-degree oracle | yes |

if it contains two oracle gates that are connected by a path. This means that there is some oracle query that depends on the answer to another oracle query. An oracle circuit that does not make adaptive oracle queries makes ***nonadaptive*** oracle queries. A ***layered*** oracle circuit is one where the oracle gates are divided into $d$ layers, and an oracle gate in layer $i$ can only be connected to oracle gates in layers $i+1, \ldots, d$, i.e., queries can depend only on answers corresponding to oracle gates in smaller layers. We can view any oracle circuit as layered: a circuit makes nonadaptive oracle queries can be viewed as a circuit with one layer of oracle gates, and a circuit that makes adaptive queries as a circuit with multiple.

**Languages.** A ***language*** $\mathcal{L} \subseteq \{0, 1\}^*$ is a set of a binary strings, and a family of circuits $\{C_n\}_{n \in \mathbb{N}}$, where for every $n$, $C_n$ takes as input a binary string of length $n$ and outputs a single bit, ***decides*** a language $\mathcal{L}$ if for every $n$ and every $x \in \{0, 1\}^n$, we have $x \in \mathcal{L}$ if and only if $C(x) = 1$. A language is a formal description of a problem; if a circuit family decides a language, we can also think of it as solving the corresponding problem.

## 3. Doubly-Efficient Interactive Proofs for Oracle Computation

In this section, we formally define our model of doubly-efficient interactive proofs for oracle computation.

First, an interactive proof is an interactive protocol between a weak verifier algorithm $\mathcal{V}$ and a powerful prover algorithm $\mathcal{P}$, where $\mathcal{P}$ tries to convince $\mathcal{V}$ of a statement of the form "$x \in \mathcal{L}$."

*Definition* 3.1 (Interactive proof system). An interactive proof for a language $\mathcal{L}$ is an interactive protocol between a probabilistic polynomial-time verifier algorithm $\mathcal{V}$ and a computationally unbounded prover algorithm $\mathcal{P}$. On common input $x$, $\mathcal{V}$ and $\mathcal{P}$ back-and-forth exchange messages in a number of rounds. In each round, $\mathcal{V}$ sends $\mathcal{P}$ a message and then $\mathcal{P}$ sends $\mathcal{V}$ a message. Both $\mathcal{V}$'s and $\mathcal{P}$'s messages can depend on $x$ and any prior messages, and $\mathcal{V}$'s messages can additionally depend on $\mathcal{V}$'s random bits $r$. At the end of this interaction, their messages form a transcript

$t = (\mathcal{V}(r), \mathcal{P})(x)$, and based on $t$, $r$, and $x$, $\mathcal{V}$ decides to accept or reject. The protocol satisfies completeness and soundness properties, namely,

- (Completeness.) For every $x \in \mathcal{L}$, $\Pr_r[V(x, t, r) = 1] \geq 2/3$.

- (Soundness.) For every $x \notin \mathcal{L}$ and for every (computationally unbounded) prover algorithm $\widetilde{\mathcal{P}}$, $\Pr_r[V(x, \widetilde{t}, r) = 1] \leq 1/3$, where $\widetilde{t} = (\mathcal{V}(r), \widetilde{\mathcal{P}})(x)$.

An interactive ***argument system*** is defined the same as an interactive proof system except that the soundness guarantee is only required to hold against provers that run in polynomial time. The soundness guarantee that a proof system is required to satisfy is called ***statistical soundness***; argument systems are only required to satisfy ***computational soundness***.

In a standard interactive proof, the verifier is constrained to run in time polynomial in $|x| = n$ and the prover is computationally unbounded. A ***doubly-efficient*** interactive proof is one where the honest prover is constrained to run in polynomial time (though soundness still holds against computationally unbounded dishonest provers), and the verifier is even more efficient, running in near-linear time. In our setting of doubly-efficient interactive proofs for oracle computation, we additionally require that the verifier makes only a *sublinear* queries to the oracle.

**A note on computational vs. statistical soundness.** Although the honest prover in all of our protocols runs in polynomial time, we give both protocols satisfying computational soundness, which are secure against any malicious polynomial-time prover, and protocols satisfying statistical soundness, which are secure even against malicious computationally unbounded provers. We stress that the weaker computational soundness guarantee is sufficient for AI safety applications, as AI models are themselves efficient algorithms.

In computationally sound protocols, there is a security parameter $\lambda$ that represents a tradeoff between the efficiency and security of the protocol. In our efficiency analysis of

computationally sound protocols, we ignore $\text{poly}(\lambda)$ factors by using the notation $O_\lambda(\cdot)$, because such factors are determined by the specific instantiation of the underlying cryptographic schemes, and it is not our goal to fully resolve them in this work. We do not think our protocols should be implemented directly in practice, but they should guide the design of practical protocols. This mirrors the development of efficient proof systems from theory to practice: first theoretical works explore possible efficiency guarantees (but pay less attention to $\text{poly}(\lambda)$ factors) (Kilian, 1992; Micali, 2000), and then applied works build practical protocols (Ben-Sasson et al., 2014). We view our result as laying the foundation towards practical protocols.

# 4. Doubly-Efficient Interactive Proofs for Robust Oracle Circuits

Here, we present our doubly-efficient interactive proof and argument systems for robust oracle circuits, which are circuits where the output remains the same even if a small fraction of the oracle answers are incorrect. To start, we give a more formal definition of what it means for a circuit to robust. In Section C.1, we present a proof system for robust circuits that make nonadaptive oracle queries. In Section C.2, we extend this to a proof system for general robust circuits (that may make adaptive queries). In Section C.3, we give an argument system for general robust circuits.

**Robustness.** Recall that what we want from this definition is that the output of a robust circuit should not change if at most a small fraction of the answers to its oracle queries are incorrect. Consider an $n$-input circuit $C$ with access to oracle $O$; we denote this by $C^O$. On input $x \in \{0,1\}^n$, $C^O$ makes a sequence of oracle queries and gets the corresponding answers, which we can denote by the string $Q = Q(x) = ((x_1, y_1), \ldots, (x_\ell, y_\ell))$, where for each $i$, $x_i$ is the $i$th oracle query and $y_i = O(x_i)$ the $i$th oracle answer. We call $Q$ the **true query-answer** string. One could also imagine substituting $Q$ with another length-$\ell$ string $Q' = ((x_1', y_1'), \ldots, (x_\ell', y_\ell'))$, where for all $i$, $x_i'$ is a binary string and $y_i'$ is a bit, though it may not be the case that $y_i' = O(x_i')$—we call a string of this form a **query-answer** string. Then we could run the computation of $C$ on $x$ with "help" from $Q'$ instead, i.e., replacing the oracle queries and answers as given by $Q$ with those given by $Q'$. We will denote this circuit by $C_{Q'}$. Note that $C^O(x) = C_Q(x)$.

*Definition* 4.1 (Closeness). Consider the following "$\varepsilon$-closeness" algorithm: given a circuit $C^O$ with $d$ layers of $\ell$ oracle gates, an input $x$, and the true query-answer string $Q = (Q_1, \ldots, Q_d)$, we set $T_1 = Q_1$ and modify some positions in $T_1$, obtaining a string $Q_1'$. Then for $i = 2, \ldots, d$, we set $T_i^{\text{in}} = C_{i-1}(x, Q_1', \ldots, Q_{i-1}')$ and $T_i = (T_i^{\text{in}}, O(T_i^{\text{in}}))$, where $O(T_i^{\text{in}})$ is obtained by applying $O$ to each query in $T_i^{\text{in}}$, and modify some positions in $T_i$, obtaining a string $Q_i'$.

If we can obtain $Q' = (Q_1', \ldots, Q_d')$ via this algorithm by modifying at most $d\varepsilon\ell$ positions (across all of the $T_i$'s), then we say that $Q'$ is $\varepsilon$-close to $Q$.[2]

*Definition* 4.2 (Robustness). We say that $C^O$ is $\varepsilon$-robust if, for any length-$n$ binary string $x$, $C_{Q'}(x) = C^O(x)$ for any $Q'$ that is $\varepsilon$-close to the true query-answer string $Q$.

## 4.1. A proof system for circuits making nonadaptive oracle queries

As a warm up, we first consider the setting where the robust circuit makes nonadaptive oracle queries.

*Theorem* 4.1 (Informal, see Theorem C.1). For any language $\mathcal{L}$ that has $\varepsilon$-robust oracle circuits making nonadaptive oracle queries with $\ell = \ell(n)$ oracle gates, depth $D = D(n) \geq \log(n)$, and size $S = S(n) \geq n$, there is a doubly-efficient interactive proof system for $\mathcal{L}$ with the following efficiency guarantees:

- The prover runs in time polynomial in $S$.

- The total communication is $(\varepsilon \cdot \ell + 1) \cdot D \cdot \text{polylog}(S)$ bits.

- The verifier runs in time $(\varepsilon \cdot n \cdot \ell + n + O(1/\varepsilon)) \cdot \text{poly}(D, \log(S)) + (1/\varepsilon) \cdot \text{polylog}(\ell)$ and makes $O(1/\varepsilon)$ oracle queries.

The proof and the protocol achieving this theorem are given in Section C.1. We highlight some key ideas here. This protocol makes key use of a tool called an *interactive proof of proximity* (IPP), which is a doubly-efficient interactive proof where the verifier is extremely efficient, running in only sublinear time, and the soundness requirement is weakened so that the verifier is only required to reject inputs that are "far" from the language with high probability (for, e.g., a fractional Hamming distance notion of distance). The high-level idea of the protocol is: given an input $x$ and an oracle circuit $C^O$, the prover claims that $C^O(x) = 1$, and the prover and the verifier do an IPP on input $(x, Q)$, where $Q$ is the true query-answer string, to show that $C_Q(x) = 1$. The problem we encounter is that in an IPP, the verifier is assumed to have reliable access to the input, whereas here the verifier must ask the unreliable prover to send bits of $Q$. We handle this by having the prover also prove that the bits she sent are correct.

---

[2]Note that, when the circuit makes nonadaptive queries, this definition of closeness is exactly a fractional Hamming distance definition, i.e., $Q$ and $Q'$ are $\varepsilon$-close if we can turn $Q$ into $Q'$ by changing at most $\varepsilon\ell$ of its positions. We use this more involved definition because in circuits that make adaptive queries, changing the output of an oracle gate can affect what the "correct" input is to another oracle gate.

## 4.2. A proof system for circuits making adaptive oracle queries

We now consider the general setting where the robust circuit can make adaptive oracle queries (which may depend on the answers to previous oracle queries).

*Theorem* 4.2 (Informal, see Theorem C.2). For any language $\mathcal{L}$ that has $\varepsilon$-robust oracle circuits with $d = d(n)$ layers of oracle gates, $\ell = \ell(n)$ oracle gates in each layer, depth $D = D(n) \geq \log(n)$, and size $S = S(n) \geq n$, there is a doubly-efficient interactive proof system for $\mathcal{L}$ with the following efficiency guarantees:

- The prover runs in time polynomial in $S$.

- The total communication is $\varepsilon DS \text{polylog}(S)$ bits.

- The verifier runs in time $\text{poly}(d, D, \log S)(\varepsilon S + 1/\varepsilon)$ and makes $\text{poly}(d)/\varepsilon$ oracle queries.

The proof and the protocol achieving this theorem are given in Section C.2. The main idea is to combine the protocol of Theorem 4.1 with recursion. In this setting, the protocol of Theorem 4.1 no longer works directly because when the verifier asks the prover for bits of the true query-answer string $Q$ in later layers, it might no longer be efficient for the prover to prove that the bits she sent are correct (since these bits may depend not just on the input $x$ but also on bits from previous layers of oracle gates). To handle this, after the prover sends bits corresponding to layer $i$ of the oracle gates, the verifier asks the prover to prove to him using an IPP that the bits she sent are computed correctly from $x$ and previous layers of oracle gates. So that the prover cannot adaptively choose what bits to send to the verifier based on the verifier's queries, we also have the prover commit to all bits in $Q$ at the start of the protocol using a universal hash function.

## 4.3. An argument system for circuits making adaptive oracle queries

We also design an interactive argument system for robust circuits that can make adaptive oracle queries, which achieves better efficiency than the previous protocols by relaxing the soundness guarantee to computational soundness.

*Theorem* 4.3 (Informal, see Theorem C.3). For any language $\mathcal{L}$ that has $\varepsilon$-robust oracle circuits of polynomial size $S$, under standard cryptographic assumptions, there is a doubly-efficient interactive argument system for $\mathcal{L}$ with the following efficiency guarantees:

- The prover runs in time polynomial in $S$.

- The total communication is $O_\lambda(1/\varepsilon)$.

- The verifier runs in time $O_\lambda(1/\varepsilon + n)$ and makes $O(1/\varepsilon)$ oracle queries.

Here $O_\lambda(\cdot)$ hides factors polynomial in the security parameter $\lambda$, which represents a tradeoff between efficiency and soundness.

Our protocol builds upon two cryptographic schemes: a commitment and a succinct argument of knowledge. Here is an overview of these schemes:

- **Commitment.** A commitment binds a sender to a message: it allows a sender to "lock" a message $m$ in a lockbox (com), so that later on, when the sender opens the lockbox, he must open it to $m$ and not something else. It is crucial that, after committing, the sender cannot produce openings to two different messages. The magic of cryptography allows that:

  - **Partial Opening.** The sender can open only partial information about the message $m$, by sending an opening. For example, if $m$ is a vector $(m_1, m_2, \ldots, m_n)$, the sender can open only $m_i$; if $m$ is a polynomial $p(x)$, the sender can open only $p(x_0)$ for some $x_0$.

  - **Succinctness.** The commitment com is much shorter than the message $m$. Furthermore, the size of the openings only depends on the size of the partial information revealed, not the size of the entire message.

- **Succinct Argument of Knowledge.** A succinct argument of knowledge allows a prover to convince that verifier that it "knows" the solution to a problem. It satisfies:

  - **Extremely Efficient Verification**. The running time of the verifier is $O_\lambda(1)$.

  - **Knowledge Extraction**. By calling the prover multiple times it is feasible to obtain the solution.

Recall that we let $Q = ((x_1, y_1), \ldots, (x_\ell, y_\ell))$ be the true query-answer string. A high-level description of our protocol is given below.

---

**Protocol** $\Pi^O_{\text{rob-arg}}(x; \varepsilon, \lambda)$ **(Interactive Argument for $\varepsilon$-robust Oracle Computation).**

**Public:** input $x$, robustness parameter $\varepsilon$, security parameter $\lambda$, oracle access to $O$.
**Tools:** commitment scheme $\text{Commit}(\cdot)$ with local opening; succinct argument of knowledge system SAoK.
**Parameters:** let $\ell$ be the number of oracle query-answer pairs in the computation. Set the number of spot-checks $k \leftarrow \Theta_\lambda(1/\varepsilon)$.

---

1. **Commitment.**

   (a) $P$ computes a transcript of query-answer pairs $Q = \langle (x_1, y_1), \ldots, (x_\ell, y_\ell) \rangle$, intended to correspond to an accepting execution.

   (b) $P$ sends $c \leftarrow \mathsf{Commit}(Q)$ to $V$.

2. **Succinct argument of knowledge.**

   (a) $P$ runs $\mathsf{SAoK}$ to prove knowledge of $Q$ that:

   $$c = \mathsf{Commit}(Q) \ \wedge \ C^Q(x) = 1,$$

   where $C^Q(x) = 1$ asserts that $Q$ is a valid query-generation transcript (i.e., each $x_i$ is consistent with the computation given prior answers), and $C$ outputs 1 given input $x$ and queries $Q$. $P$ sends the succinct proof $\pi_{\mathsf{SAoK}}$ to $V$.

   (b) $V$ verifies $\pi_{\mathsf{SAoK}}$; if verification fails then reject, else accept.

3. **Random oracle spot-checks.**

   (a) $V$ samples a random set of indices $I \subseteq [\ell]$ with $|I| = k$ and sends $I$ to $P$.

   (b) For each $i \in I$, $P$ opens every $(x_i, y_i)$ under commitment $c$.

   (c) For each $i \in I$, $V$ verifies the opening and checks $y_i = O(x_i)$. If any check fails, $V$ rejects.

**Output:** $V$ accepts iff all checks in (2) and (3) pass.

---

At a high level, to see why the protocol works, the succinct argument of knowledge ensures the existence of $Q$ that makes the computation accept, while being consistent with the commitment $c$. The random spot-checks ensure that, except with negligible probability, $Q$ has no more than $\varepsilon \ell$ incorrect oracle query-answer pairs. We refer readers to Section C.3 for the full protocol and analysis.

## 5. A Doubly-Efficient Interactive Argument for Low-Degree Oracles

In this section, we present a doubly-efficient argument system for circuits with access to a low-degree oracle.

First, we say that an oracle $O$ is *low-degree* if its behavior on inputs of length up to $n$ can be captured by a polynomial-degree polynomial over a finite field.

*Definition* 5.1 (Degree of Oracle). Let $O : \{0, 1\}^* \to \{0, 1\}$ be an oracle and $O_n$ be the restriction of $O$ to inputs of length at most $n$. We say that $O$ has degree $d(n)$ if for every $n \in \mathbb{N}$, there exists a degree-$d(n)$ polynomial $P_n$ over some finite field $\mathbb{F}$ of size $2^{O(n)}$, such that we can encode every

$x \in \{0, 1\}^{\leq n}$ into $\mathbb{F}$ and $O_n(x) = P_n(x)$ for every such $x$.

Our result is as follows.

*Theorem* 5.1 (Informal, see Theorem D.1). For any language $\mathcal{L}$ that has low-degree oracle circuits of polynomial size $S$, under standard cryptographic assumptions, there is a doubly-efficient interactive argument system for $\mathcal{L}$ with the following efficiency guarantees:

- The prover runs in polynomial time in $S$;

- The total communication is $O_\lambda(1)$;

- The verifier runs in time $O_\lambda(n)$ and makes 1 oracle query.

Here $O_\lambda(\cdot)$ hides factors polynomial in the security parameter $\lambda$, which is the tradeoff parameter between efficiency and soundness.

Our protocol builds on succinct argument of knowledge and polynomial commitment schemes. We refer readers to **??** for the full protocol and analysis. We note that one extraordinary feature of our protocol is that the verifier only needs to make one query to the oracle, regardless of the number of oracle queries made by the computation.

## 6. Conclusion

In this paper, we initiated the study of single-prover interactive proofs for oracle-aided computations as an approach to scalable AI safety. While there do not exist interactive proofs for all oracle-aided computations, we presented relativizing doubly-efficient single-prover interactive proofs for two natural settings: (1) where the computation is robust and (2) where the oracle is a low-degree polynomial. We end on some interesting further directions of study.

- **Experimental validation.** One important direction for future work is experimental validation of our protocols. For example, it would be interesting to study which realistic scalable oversight tasks satisfy our robustness assumption and whether our protocols can be implemented efficiently in practice.

- **Other settings.** In this work, we considered the settings of robust computation and low-degree oracles. Are there other natural assumptions that one could make on the computation or on the oracle that would allow us to obtain relativizing single-prover interactive proofs? For example, it would be interesting to consider a broader class of "learnable" oracles.

- **Scope of robust computation.** As mentioned previously, many natural tasks may be able to be made

robust using redundancy. Can we formalize this intuition as some theoretical model (and then apply our protocol for robust computation)?

- **Verifying tasks that we don't know how to do.** In our model the verifier has access to the computation that he wants the prover to compute on some input; he's just too efficient to perform the computation by himself. It would be interesting to study a model where not only can the verifier not perform the computation by himself, but he does not even have access to a description of the computation—this would, for example, capture the problem of scalable oversight in the setting where we are not able to perform the task that we are training the AI to do. Could we construct protocols for this setting?

## Acknowledgments

The authors would like to acknowledge support provided by the UK AISI Alignment Project. Zoe Xi is supported by an Akamai Presidential Fellowship.

## Impact Statement

This paper introduces formal approaches to enhancing factuality and safety in AI architectures, particularly Large Language Models (LLMs). This represents a crucial area of machine learning research with significant practical implications. However, there is a significant gap between our theoretical understanding of these issues and real world practice, as current methods continue to rely heavily on expensive human feedback, which substantially limits their scalability.

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

# A. Organization of Appendices

These appendices contain the technical parts of this work. In Section B we provide the rest of our preliminaries, and in Section C and Section D we provide our results in full for the settings of robust computation and low-degree oracles, respectively.

# B. Preliminaries

## B.1. Interactive proofs

An interactive proof is an interactive protocol between a weak verifier algorithm $\mathcal{V}$ and a powerful prover algorithm $\mathcal{P}$, where $\mathcal{P}$ tries to convince $\mathcal{V}$ of a statement of the form "$x \in \mathcal{L}$."

*Definition* B.1 (Interactive proof system). An interactive proof for a language $\mathcal{L}$ is an interactive protocol between a probabilistic polynomial-time verifier algorithm $\mathcal{V}$ and a computationally unbounded prover algorithm $\mathcal{P}$. On common input $x$, $\mathcal{V}$ and $\mathcal{P}$ back-and-forth exchange messages in a number of rounds. In each round, $\mathcal{V}$ sends $\mathcal{P}$ a message and then $\mathcal{P}$ sends $\mathcal{V}$ a message. Both $\mathcal{V}$'s and $\mathcal{P}$'s messages can depend on $x$ and any prior messages, and $\mathcal{V}$'s messages can additionally depend on $\mathcal{V}$'s random bits $r$. At the end of this interaction, their messages form a transcript $t = (\mathcal{V}(r), \mathcal{P})(x)$, and based on $t$, $r$, and $x$, $\mathcal{V}$ decides to accept or reject. The protocol satisfies completeness and soundness properties, namely,

- (Completeness.) For every $x \in \mathcal{L}$, $\Pr_r[V(x, t, r) = 1] \geq 2/3$.

- (Soundness.) For every $x \notin \mathcal{L}$ and for every (computationally unbounded) prover algorithm $\widetilde{P}$, $\Pr_r[V(x, \widetilde{t}, r) = 1] \leq 1/3$, where $\widetilde{t} = (\mathcal{V}(r), \widetilde{\mathcal{P}})(x)$.

An interactive **argument system** is defined the same as an interactive proof system except that the soundness guarantee is only required to hold against provers that run in polynomial time. The soundness guarantee that a proof system is required to satisfy is called **statistical soundness**; argument systems are only required to satisfy **computational soundness**.

In a standard interactive proof, the verifier is constrained to run in time polynomial in $|x| = n$ and the prover is computationally unbounded. A **doubly-efficient** interactive proof is one where the honest prover is constrained to run in polynomial time (though soundness still holds against computationally unbounded dishonest provers), and the verifier is even more efficient, running in near-linear time. In our setting of doubly-efficient interactive proofs for oracle computation, we additionally require that the verifier makes only a *sublinear* queries to the oracle.

We will make use of a construction of doubly-efficient interactive proofs due to Goldwasser, Kalai, and Rothblum:

*Theorem* B.1 ((Goldwasser et al., 2015)). Let $\mathcal{L}$ be a language that has logspace-uniform Boolean circuits of depth $D = D(n)$ and space $S = S(n)$. There is a doubly-efficient public-coin interactive proof for $\mathcal{L}$ with the following parameters:

- soundness error $1/2$;[3]

- communication complexity $D \cdot \text{polylog}(S)$;

- verifier running time $n \cdot \text{poly}(D, \log S)$; and

- prover running time $\text{poly}(S)$.

## B.2. Interactive proofs of proximity

Our proof systems for robust circuits make key use of an interactive proof of proximity (IPP). Loosely speaking, an IPP is a doubly-efficient interactive proof where the verifier is extremely efficient, running in only sublinear time, and the soundness requirement is weakened so that the verifier is only required to reject inputs that are "far" from the language with high probability, for, e.g., a fractional Hamming distance notion of distance. We will deal with IPPs for **pair languages**, where the input to the verifier is a pair $(x, Q)$ consisting of an **explicit input** $x$ that the verifier has direct access to and an **implicit input** $Q$ that the verifier has query access to. We say that $(x, Q)$ is $\varepsilon$-**Hamming-far** from pair language $\mathcal{L}$ for **proximity parameter** $\varepsilon \in (0, 1]$ if $Q$ differs in at least an $\varepsilon$ fraction of its positions from every $Q'$ such that $(x, Q') \in \mathcal{L}$, and that it is $\varepsilon$-**Hamming-close** otherwise.

---

[3]Note that we can boost soundness using parallel repetition.

*Definition* B.2 (Interactive proof of proximity (Rothblum et al., 2013; Rothblum & Rothblum, 2020)). An interactive proof of proximity for a pair language $\mathcal{L}$ is an interactive protocol between a probabilistic sublinear-time verifier algorithm $\mathcal{V}$ and a polynomial-time prover algorithm $\mathcal{P}$. Both $\mathcal{V}$ and $\mathcal{P}$ have access to explicit input $x$ and a proximity parameter $\varepsilon$, and $\mathcal{V}$ has query access to implicit input $Q$ while $\mathcal{P}$ has direct access to $Q$. The protocol satisfies completeness and a relaxed notion of soundness, namely

- (Completeness.) For every $(x, Q) \in \mathcal{L}$ and proximity parameter $\varepsilon \in (0, 1]$, we have $\Pr_r[\mathcal{V}^Q(x, t) = 1] = 1$, where $r$ denotes $\mathcal{V}$'s random bits and $t = (\mathcal{V}^Q(r), \mathcal{P}(Q))(x, |Q|, \varepsilon)$ is the transcript of the protocol.

- (Soundness.) For every $(x, Q)$ that is $\varepsilon$-Hamming-far from $\mathcal{L}$, and for every (computationally unbounded) prover algorithm $\widetilde{\mathcal{P}}$, we have $\Pr_r[\mathcal{V}^Q(x, \widetilde{t}) = 1] \leq 1/2$, where $r$ denotes $\mathcal{V}$'s random bits and $\widetilde{t} = (\mathcal{V}^Q(r), \widetilde{\mathcal{P}}(Q))(x, |Q|, \varepsilon)$ is the transcript.

The best-known IPP is due to Rothblum and Rothblum:

*Theorem* B.2 ((Rothblum & Rothblum, 2020)). Let $\varepsilon = \varepsilon(n) \in (0, 1]$ be a proximity parameter, and let $\mathcal{L}$ be a pair language that is computable by logspace-uniform Boolean circuits of depth $D = D(n) \geq \log(n)$ and size $S = S(n) \geq n$ with fan-in 2, where we use $n$ for the length of the explicit input and $m$ for the length of the implicit input. Then there is a public-coin IPP with $\varepsilon$ proximity for $\mathcal{L}$ with the following parameters:

- soundness error $1/2$;

- query complexity $q = O(1/\varepsilon)$;

- communication complexity $cc = \varepsilon \cdot m \cdot D \cdot \mathrm{polylog}(S)$;

- round complexity $D \cdot \mathrm{polylog}(S)$;

- verifier running time $\varepsilon \cdot n \cdot m \cdot \mathrm{poly}(D, \log(S)) + (1/\varepsilon) \cdot \mathrm{polylog}(m)$; and

- prover running time $\mathrm{poly}(S)$.

Furthermore, the verifier can make all of his queries to the input at the end of the interaction.

The IPPs that we use in our proof systems actually *differ* from standard IPPs in that the verifier does not have direct query access to the implicit input; instead, the verifier asks the prover for positions in the input, and the prover may lie. However, it will be the case that with good probability the verifier will accept only if the prover sends the positions consistent with a "fixed" string that is close to the input, so that in some sense we are able to recover the standard IPP setting. We will write that the prover answers the verifier's queries to indices $J$ of the implicit input ***consistent*** with a string $Q$ if the prover's answer is $Q|_J$.

### B.3. Universal hash functions

Our proof system for adaptive robust circuits makes use of universal hash functions:

*Definition* B.3 (Universal hash function). An $\varepsilon$-universal hash function $H : K \times M \to T$ is an algorithm with the following guarantee: for all $m_0, m_1 \in M$, $\Pr[H(k, m_0) = H(k, m_1)] \leq \varepsilon$, where the probability is over the random choice of $k \in K$.

There is a well-known construction of a universal hash function using polynomials.

*Lemma* B.4. For every $M, T \in \mathbb{N}$, there is a family of $1/|T|$-universal hash function family $H$, where the hash key $k \leftarrow K$ can be represented using $\max\{M, T\} + T$ random bits.

### B.4. Arguments of knowledge

Our argument systems make use of arguments of knowledge, which intuitively are arguments wherein a prover establishes not only that their claim is true but that they know a witness.

Let $\mathcal{R} \in \{0, 1\}^* \times \{0, 1\}^* \times \{0, 1\}^*$ be ternary relation. If $(\mathsf{pp}, x, w) \in \mathcal{R}$, we say that $\mathsf{pp}$ are the public parameters, $x$ is a statement and $w$ is a witness for $x$.

*Definition* B.5 (Interactive argument system). Let $\ell \geq 0$ be an integer. A $(2\ell + 1)$-message public-coin argument system $\Pi = (\mathsf{Setup}, \mathcal{P}, \mathcal{V})$ for a relation $\mathcal{R}$ consists of a PPT (probabilistic polynomial time) algorithm $\mathsf{Setup}$ and a $(2\ell + 1)$-message protocol between an interactive PPT prover $\mathcal{P}$ and an interactive PPT verifier $\mathcal{V}$ associated with a tuple $(X, W, (Z_{i-1}, C_i)_{i \in [\ell]}, Z_\ell)$, with the following properties:

- The $\mathsf{Setup}$ algorithm takes as input the security parameter $1^\lambda$ and outputs some public parameters $\mathsf{pp}$.

- Both $\mathcal{P}$ and $\mathcal{V}$ receive as input the public parameters $\mathsf{pp}$ and a statement $x_0 = x \in X$. The prover $\mathcal{P}$ additionally receives a witness $w_0 = w \in W$.

- The public parameters $\mathsf{pp}$, the statement $x_0$, and the $2\ell + 1$ messages sent by $\mathcal{P}$ and $\mathcal{V}$ in the protocol, are collectively called a transcript, labelled as

$$(\mathsf{pp}, x_0, z_0, c_1, \ldots, z_{\ell-1}, c_\ell, z_\ell),$$

where $z_i \in Z_i$ is sent by $\mathcal{P}$ and $c_i \in C_i$ is sent by $\mathcal{V}$.

- The challenges $c_i$ are sampled by $\mathcal{V}$ uniformly randomly from $C_i$.

A transcript $(\mathsf{pp}, x_0, z_0, c_1, \ldots, z_{\ell-1}, c_\ell, z_\ell)$ is said to be accepting for $\Pi$ if $\mathcal{V}(\mathsf{pp}, x_0, z_0, c_1, \ldots, z_{\ell-1}, c_\ell, z_\ell) = 1$ holds.

Now we define completeness and knowledge soundness:

*Definition* B.6 (Completeness). An argument system $\Pi = (\mathsf{Setup}, \mathcal{P}, \mathcal{V})$ for the relation $\mathcal{R}$ has statistical completeness with correctness error $\epsilon$ if for all adversaries $\mathcal{A}$,

$$\Pr \left[ b = 0 \wedge (\mathsf{pp}, x, w) \in \mathcal{R} \,\middle|\, \begin{array}{c} \mathsf{pp} \leftarrow \mathsf{Setup}(1^\lambda) \\ (x, w) \leftarrow \mathcal{A}(\mathsf{pp}) \\ (\mathsf{tr}, b) \leftarrow \langle \mathcal{P}(\mathsf{pp}, x, w), \mathcal{V}(\mathsf{pp}, x) \rangle \end{array} \right] \leq \epsilon(\lambda).$$

Furthermore, we say that $\Pi$ satisfies perfect completeness if $\epsilon = 0$.

*Definition* B.7 (Knowledge soundness). An argument system $\Pi = (\mathsf{Setup}, \mathcal{P}, \mathcal{V})$ is knowledge sound with knowledge error $\kappa$ for the relation $\mathcal{R}^*$ if there exists an expected PPT extractor $\mathcal{E}$ such that for any stateful PPT adversary $\mathcal{P}^*$:

$$\Pr \left[ b = 1 \wedge (\mathsf{pp}, x, w) \notin \mathcal{R}^* \,\middle|\, \begin{array}{c} \mathsf{pp} \leftarrow \mathsf{Setup}(1^\lambda) \\ (x, \mathsf{st}) \leftarrow \mathcal{P}^*(\mathsf{pp}) \\ (\mathsf{tr}, b) \leftarrow \langle \mathcal{P}^*(\mathsf{pp}, x, \mathsf{st}), \mathcal{V}(\mathsf{pp}, x) \rangle \\ w \leftarrow \mathcal{E}_{\mathcal{P}^*}(\mathsf{pp}, x) \end{array} \right] \leq \kappa(\lambda).$$

Here, the extractor $\mathcal{E}$ has a black-box oracle access to the (malicious) prover $\mathcal{P}^*$ and can rewind it to any point in the interaction.

The classical Kilian's protocol (Kilian, 1992) is an argument of knowledge for any NP language, i.e., $\mathcal{R} = (\bot, x, w)$ where $(x, w)$ is a valid instance-witness pairs for some NP relation. Furthermore, Kilian's protocol has an additional property, called succinctness: the communication complexity and the running time of the verifier are $\mathsf{poly}(\lambda)$.

*Theorem* B.3. Assuming the existence of collision-resistant hash functions, there exists a succinct argument of knowledge for any NP relation $\mathcal{R}$ with knowledge error negligible in the security parameter $\lambda$.

## B.5. Hash trees

Our argument system for robust oracle circuits makes use of a hash tree, which is a verifiable hash function that supports local openings (i.e., it allows a prover to commit to a list of elements and later open any individual element) that can be instantiated from any collision-resistant hash function (Merkle, 1990).

*Definition* B.8. A hash tree consists of a tuple of algorithms $(\mathcal{G}_{\mathsf{ht}}, \mathcal{H}_{\mathsf{ht}}, \mathcal{P}_{\mathsf{ht}}, \mathcal{V}_{\mathsf{ht}})$ with the following syntax:

- $\mathcal{G}_{\mathsf{ht}}(1^\lambda, S) \to \mathsf{hk}$. This is a randomized algorithm that takes as input a security parameter $\lambda$, and a space bound $S$, and outputs a hash key $\mathsf{hk}$. We implicitly assume that $\mathsf{hk}$ includes $1^\lambda, S$.

- $\mathcal{H}_{\mathsf{ht}}(\mathsf{hk}, D) \to \mathsf{rt}$. This is a deterministic algorithm that on input a hash key $\mathsf{hk}$ and a database $D \in \{0, 1\}^S$ outputs a root $\mathsf{rt}$.

- $\mathcal{P}_{\mathsf{ht}}(\mathsf{hk}, D, i) \to \rho$. This is a deterministic algorithm that on input a hash key $\mathsf{hk}$, a database $D \in \{0,1\}^S$, an index $i \in [S]$, outputs an opening proof $\rho$. When a set $I \subseteq [S]$ is given as input, we interpret this as providing proofs for $(i, j)$ for all $j \in [m]$.

- $\mathcal{V}_{\mathsf{ht}}(\mathsf{hk}, \mathsf{rt}, i, v, \rho) \to b$. This is a deterministic algorithm that on input a hash key $\mathsf{hk}$, a hash tree root $\mathsf{rt}$, an index $i \in [S]$, a bit $v \in \{0,1\}$, and an opening proof $\rho$, outputs a bit $b \in \{0,1\}$ indicating whether to accept or reject the opening proof. When a set $I \subseteq [S]$ and value $v \in \{0,1\}^{|I|}$ is given as input, we interpret this as verifying proofs for $v[j]$ and index $i$ for all $j \in [|I|]$ and $i \in I$.

We require $(\mathcal{G}_{\mathsf{ht}}, \mathcal{H}_{\mathsf{ht}}, \mathcal{P}_{\mathsf{ht}}, \mathcal{V}_{\mathsf{ht}})$ to satisfy the following properties:

- **Opening completeness.** For any $\lambda, S \in \mathbb{N}$, $i \in [S]$, database $D \in \{0,1\}^S$, it holds that

$$
\Pr\left[ \mathcal{V}_{\mathsf{ht}}(\mathsf{hk}, \mathsf{rt}, i, D[i], \rho) = 1 \;:\; \begin{array}{l} \mathsf{hk} \leftarrow \mathcal{G}_{\mathsf{ht}}(1^\lambda, S) \\ \mathsf{rt} = \mathcal{H}_{\mathsf{ht}}(\mathsf{hk}, D) \\ \rho = \mathcal{P}_{\mathsf{ht}}(\mathsf{hk}, D, i) \end{array} \right] = 1.
$$

- **Efficiency.** In the opening completeness and extraction correctness experiment above for $S \leq 2^\lambda$, $\mathcal{G}_{\mathsf{ht}}$ and $\mathcal{V}_{\mathsf{ht}}$ run in $\mathsf{poly}(\lambda)$ time, and $\mathcal{H}_{\mathsf{ht}}$ and $\mathcal{P}_{\mathsf{ht}}$ run in $S \cdot \mathsf{poly}(\lambda)$ time. Also we require $|\mathsf{hk}| \leq \mathsf{poly}(\lambda)$ and $|\rho| \leq \mathsf{poly}(\lambda)$.

- **Binding.** For any non-uniform polynomial-time algorithm $A = \{A_\lambda\}_{\lambda \in \mathbb{N}}$ and polynomial $S$, there exists a negligible function $\mu$ such that for all $\lambda \in \mathbb{N}$, it holds that

$$
\Pr\left[ \begin{array}{l} v \neq v' \\ \wedge\, 1 = \mathcal{V}_{\mathsf{ht}}(\mathsf{hk}, \mathsf{rt}, i, v, \rho) \\ \wedge\, 1 = \mathcal{V}_{\mathsf{ht}}(\mathsf{hk}, \mathsf{rt}, i, v', \rho') \end{array} \;:\; \begin{array}{l} \mathsf{hk} \leftarrow \mathcal{G}_{\mathsf{ht}}(1^\lambda, S(\lambda)) \\ (\mathsf{rt}, i, v, v', \rho, \rho') \leftarrow A_\lambda(\mathsf{hk}) \end{array} \right] \leq \mu(\lambda).
$$

*Remark* B.9. For convenience, we may take the characters in the database $D$ as some finite set $\Sigma$ instead of just binary bits. The definition and properties of hash trees can be easily adapted to this setting, with a $\log |\Sigma|$ overhead on all efficiency parameters.

### B.6. Polynomial commitments

Our argument system for the setting where the oracle is low-degree uses a special commitment scheme called a polynomial commitment, which allows a prover to commit to a degree-bounded polynomial and later open evaluations of the polynomial at specific points, along with (interactive) proofs that the evaluations are correct.

*Definition* B.10. A (non-interactive) commitment scheme over $\mathcal{M}$ with slack space $\mathcal{S}$ is a tuple of polynomial-time probabilistic algorithms $\mathsf{CM} = (\mathsf{Setup}, \mathsf{Commit}, \mathsf{Open})$ with the following syntax.

- $\mathsf{Setup}(1^\lambda, d) \to \mathsf{pp}$: Sample public parameters given a security parameter $\lambda$ and message length $d$.

- $\mathsf{Commit}(\mathsf{pp}, f) \to (C, \mathsf{st})$: Use the public parameters $\mathsf{pp}$ to compute a commitment $C$ to a message $f \in \mathcal{M}$ and an auxiliary state $\mathsf{st}$.

- $\mathsf{Open}(\mathsf{pp}, C, f, \mathsf{st}) \to b$: Takes public parameters $\mathsf{pp}$, a commitment $C$, a message $f \in \mathcal{M}$, and an auxiliary state $\mathsf{st}$, and outputs a bit $b$ indicating whether $C$ is a valid commitment to $f$ under $\mathsf{pp}$.

We require commitment schemes to satisfy the following completeness and (relaxed) binding properties.

*Definition* B.11 (Completeness). A commitment scheme $\mathsf{CM} = (\mathsf{Setup}, \mathsf{Commit}, \mathsf{Open})$ satisfies completeness if for all $\lambda, d \in \mathbb{N}$, and for every $f \in \mathcal{M}$

$$
\Pr\left[ \mathsf{Open}(\mathsf{pp}, C, f, \mathsf{st}, \bot) = 1 \;\middle|\; \begin{array}{l} \mathsf{pp} \leftarrow \mathsf{Setup}(1^\lambda, d) \\ (C, \mathsf{st}) \leftarrow \mathsf{Commit}(\mathsf{pp}, f) \end{array} \right] \geq 1 - \mathsf{negl}(\lambda).
$$

*Definition* B.12 (Binding). A commitment scheme $\mathsf{CM} = (\mathsf{Setup}, \mathsf{Commit}, \mathsf{Open})$ satisfies relaxed binding if for every PPT adversary $\mathcal{A}$,

$$
\Pr\left[ \begin{array}{c} f \neq f' \text{ with } f, f' \in \mathcal{M} \\ \wedge \\ \mathsf{Open}(\mathsf{pp}, C, f, \mathsf{st}) = \mathsf{Open}(\mathsf{pp}, C, f', \mathsf{st}') = 1 \end{array} \;\middle|\; \begin{array}{l} \mathsf{pp} \leftarrow \mathsf{Setup}(1^\lambda, d) \\ (C, (f, \mathsf{st}), (f', \mathsf{st}')) \leftarrow \mathcal{A}(\mathsf{pp}) \end{array} \right] \leq \mathsf{negl}(\lambda).
$$

Now we define an extractable polynomial commitment.

*Definition* B.13 (Extractable polynomial commitment scheme). Let $\mathrm{PC} = (\mathsf{Setup}_{\mathrm{CM}}, \mathsf{Commit}, \mathsf{Open}, \mathsf{Setup}_{\mathrm{IP}}, \mathcal{P}, \mathcal{V})$ be a tuple of algorithms. PC is a functional commitment scheme for function class $\mathcal{F}$ if

- $(\mathsf{Setup}_{\mathrm{CM}}, \mathsf{Commit}, \mathsf{Open})$ is a commitment scheme over the function class

$$\mathcal{M} := \mathcal{F}.$$

- $(\mathsf{Setup}_{\mathrm{IP}}, \mathcal{P}, \mathcal{V})$ is an argument system for the relation

$$(\mathsf{pp}, (\mathsf{pp}_{\mathrm{CM}}, C, \mathbf{x}, \mathbf{u}), (f, \mathsf{st})) \in \mathcal{R} \quad \Longleftrightarrow \quad \mathsf{Open}(\mathsf{pp}_{\mathrm{CM}}, C, f, \mathsf{st}) = 1 \wedge f(\mathbf{x}) = \mathbf{u}.$$

The class of functions $\mathcal{F}$ supported by a polynomial commitment scheme will be a set of polynomials. We say that the polynomial commitment scheme satisfies completeness and knowledge soundness if $(\mathsf{Setup}_{\mathrm{IP}}, \mathcal{P}, \mathcal{V})$ is complete and knowledge sound respectively.

*Theorem* B.4 (See (Cini et al., 2024)). Assuming the standard (Module)-SIS assumption, there exists an extractable polynomial commitment scheme with $\mathrm{poly}(\lambda, \log L)$ communication and verification times, where $L$ is the degree bound of the committed polynomial. Furthermore, the number of rounds of the local opening argument system is $O(\log L)$.

# C. Doubly-Efficient Interactive Proofs for Robust Oracle Circuits

Here we present our doubly-efficient single-prover interactive proof and argument systems for robust oracle circuits. We first present a proof system for robust circuits that make nonadaptive oracle queries in Section C.1 below. In Section C.2, we extend this to a proof system for general robust circuits (that may make adaptive queries). In Section C.3, we give an argument system for general robust circuits.

## C.1. A proof system for circuits making nonadaptive oracle queries

*Theorem* C.1. Let $\varepsilon = \varepsilon(n) \in (0, 1]$, and let $\mathcal{L}$ be a language that has $\varepsilon$-robust logspace-uniform oracle Boolean circuits with $\ell = \ell(n)$ oracle gates, depth $D = D(n) \geq \log(n)$, and size $S = S(n) \geq n$, where all gates except possibly the oracle gates have fan-in 2. Then there is a doubly-efficient interactive proof system for $\mathcal{L}$ with the following parameters:

- soundness error $1/2$;

- query complexity $q = O(1/\varepsilon)$;

- communication complexity $cc = (\varepsilon \cdot \ell + 1) \cdot D \cdot \mathrm{polylog}(S)$;

- round complexity $D \cdot \mathrm{polylog}(S)$;

- verifier running time $(\varepsilon \cdot n \cdot \ell + n + O(1/\varepsilon)) \cdot \mathrm{poly}(D, \log(S)) + (1/\varepsilon) \cdot \mathrm{polylog}(\ell)$; and

- prover running time $\mathrm{poly}(S)$.

*Proof.* The protocol is given in Figure 1. Here we analyze its efficiency, completeness, and soundness.

**Efficiency.** The claimed parameters follow by composing: (i) the IPP for $\mathcal{L}_{\mathrm{acc}}$ with proximity parameter $\varepsilon$ and implicit input length $m = \Theta(\ell)$ and (ii) one invocation of the DEIP on a statement of size $n + O(|J|)$. The verifier additionally makes $|J|$ oracle queries in order to check $y_i = O(x_i)$ for $i \in J$.

**Completeness.** If $C^O(x) = 1$, then $(x, Q) \in \mathcal{L}_{\mathrm{acc}}$. The prover can answer the verifier's request in step 2 with $A = Q|_J$. Then the verifier's oracle correctness checks pass, the DEIP claim $C_0(x)|_J = A^{\mathrm{in}}$ holds, and by completeness of the IPP the verifier accepts.

**Soundness.** Fix any (computationally unbounded) dishonest prover $\widetilde{\mathcal{P}}$, and suppose the outer verifier accepts.

---

**DEIP for robust, nonadaptive setting**

**Input:** The prover $\mathcal{P}$ and verifier $\mathcal{V}$ receive $x \in \{0,1\}^n$, a succinct description of an $\varepsilon$-robust logspace-uniform oracle circuit $C$ containing one oracle-gate layer of $\ell$ oracle gates, and oracle access to $O : \{0,1\}^* \to \{0,1\}$.

**Notations and tools:** Enumerate the oracle gates by $1, \ldots, \ell$. Let $Q = Q(x) = ((x_1, y_1), \ldots, (x_\ell, y_\ell))$ denote the true query-answer string and $Q^{\text{in}}$ denote the string of queries in $Q$. We split $C^O$ into two ordinary circuits with no oracle gates: $C_0$ computes $Q^{\text{in}}$ from $x$, and $C_1$ computes $C^O(x)$ from $(x, Q)$. We make use of the following tools:

- An IPP for the pair language $\mathcal{L}_{\text{acc}} = \{(x, Q) : C_1(x, Q) = 1\}$ with proximity parameter $\varepsilon$ (instantiated by Theorem B.2).

- A DEIP for the language asserting correctness of $C_0(x)$ on a queried subset of coordinates (instantiated by Theorem B.1).

**The protocol:**

1. The prover and an IPP verifier run the message-exchange portion of the IPP for $\mathcal{L}_{\text{acc}}$ with proximity parameter $\varepsilon$ on input $(x, Q)$, where $Q$ is implicit. At the end of this interaction, the IPP verifier outputs a set of indices $J \subseteq [\ell]$ that it wishes to query (here we use that in Theorem B.2, the IPP verifier can postpone all its queries to the implicit input until the end of the interaction).

2. The verifier asks the prover to send the query-answer pairs at indices $J$, the prover responds with

$$A = \{(x_i, y_i)\}_{i \in J},$$

   and the verifier sends $A$ to the IPP verifier. The verifier checks that the oracle calls in $A$ are correct: for all $i \in J$, it checks that $y_i = O(x_i)$. Letting $A^{\text{in}} = (x_i)_{i \in J}$ denote the string of queries in $A$, the prover and a DEIP verifier engage in a DEIP to prove the claim $C_0(x)|_J = A^{\text{in}}$.

3. The verifier accepts if the IPP and DEIP verifiers accept and all of its checks pass, and rejects otherwise.

*Figure 1.* DEIP for robust circuits making nonadaptive queries

Let $J$ be the set of indices output by the IPP verifier in step 1, and let $A = \{(x_i, y_i)\}_{i \in J}$ be the prover's response in step 2. Define the event

$$E := \big(\forall i \in J, \; y_i = O(x_i)\big) \; \wedge \; \big(C_0(x)|_J = A^{\mathrm{in}}\big).$$

Note that the first conjunct is checked perfectly by the verifier (using oracle access), and the second conjunct is exactly the DEIP statement.

Condition on event $E$. Since queries are nonadaptive, $C_0(x)|_J$ is the true query-answer string $(x_i)_{i \in J}$ of the oracle circuit on input $x$. Therefore $E$ implies that

$$A = Q|_J.$$

In other words, the answers that are fed to the IPP verifier in step 2 are exactly the answers that the IPP verifier would obtain if it had genuine query access to the true implicit input $Q$.

Now consider the IPP soundness guarantee for $\mathcal{L}_{\mathrm{acc}}$ with proximity parameter $\varepsilon$ (Theorem B.2). Conditioned on $E$, the IPP verifier's view is distributed exactly as in a standard IPP execution with genuine query access to implicit input $Q$. Therefore, if $(x, Q)$ is $\varepsilon$-Hamming-far from $\mathcal{L}_{\mathrm{acc}}$, then (even allowing an unbounded prover) the IPP verifier accepts with probability at most $1/2$.

Equivalently, if the outer verifier accepts with probability larger than the sum of the DEIP soundness error and $1/2$, then $(x, Q)$ must be $\varepsilon$-Hamming-close to $\mathcal{L}_{\mathrm{acc}}$. In particular, there exists a transcript $Q'$ such that

$$C_1(x, Q') = 1 \quad \text{and} \quad \Delta(Q, Q') \le \varepsilon \ell.$$

As $Q'$ is a query-answer string for input $x$, and it differs from the true query-answer string $Q$ on at most an $\varepsilon$ fraction of indices, by $\varepsilon$-robustness of $C^O$, this implies that $C^O(x) = C_1(x, Q) = C_1(x, Q') = 1$.

Finally, unconditioning $E$, the only ways the outer verifier can accept while $C^O(x) = 0$ are: (i) the DEIP accepts a false statement (which happens with probability at most its soundness error), or (ii) conditioned on $E$, the IPP accepts even though $(x, Q)$ is $\varepsilon$-far from $\mathcal{L}_{\mathrm{acc}}$ (which happens with probability at most the IPP soundness error). Using standard constant-factor repetition to set both soundness errors to at most $1/4$, we obtain overall soundness error at most $1/2$.

$\square$

## C.2. A proof system for circuits making adaptive oracle queries

*Theorem* C.2. Let $\varepsilon = \varepsilon(n) \in (0, 1]$, and let $\mathcal{L}$ be a language that has $\varepsilon$-robust logspace-uniform oracle Boolean circuits containing $d = d(n)$ oracle-gate layers each with $\ell = \ell(n)$ oracle gates, depth $D = D(n) \ge \log(n)$, and size $S = S(n) \ge n$, where all gates except possibly the oracle gates have fan-in 2. Then there is a doubly-efficient interactive proof system for $\mathcal{L}$ with the following parameters:

- soundness error $1/2$;
- query complexity $q = \mathsf{poly}(d)/\varepsilon$;
- communication complexity $cc = \varepsilon D S \mathsf{polylog}(S)$;
- round complexity $dD\mathsf{polylog}(S)$;
- verifier running time $\mathsf{poly}(d, D, \log S)(\varepsilon S + 1/\varepsilon)$; and
- prover running time $\mathsf{poly}(S)$.

*Proof.* The protocol is given in Figure 2. We analyze the protocol's completeness, soundness, and efficiency.

**Completeness.** If $x$ is in the language, then the honest prover answers the queries to the inputs in the IPPs consistent with the true query-answer string $Q$. Completeness then follows from completeness of the IPPs and the DEIP.

**Soundness.** Fix any cheating prover $\widetilde{P}$ and $x \notin \mathcal{L}$, and we show that $\mathcal{V}$ accepts with probability at most $p = 1/2$.

---

**DEIP for robust, adaptive setting**

**Input:** The prover $\mathcal{P}$ and verifier $\mathcal{V}$ receive $x \in \{0,1\}^n$, a succinct description of an $\varepsilon$-robust logspace-uniform oracle circuit $C$ containing $d$ oracle-gate layers each with $\ell$ oracle gates, and oracle access to $O : \{0,1\}^* \to \{0,1\}$.

**Parameters:** For each layer $i \in [d]$, let $\varepsilon_i$ be the proximity parameter for the corresponding IPP and $h_i$ a hash output length (set later). We assume that $\sum_{i=1}^{d} \varepsilon_i \leq \varepsilon$.

**Notations:** For each $i \in [d]$, enumerate the oracle gates in layer $i$ by $(i,1), \ldots, (i,\ell)$. For $i \in [d]$, $j \in [\ell]$, let $x_{i,j}$ denote the query made at oracle gate $(i,j)$ and $y_{i,j} = O(x_{i,j})$ the corresponding answer. Let $Q_i := ((x_{i,1}, y_{i,1}), \ldots, (x_{i,\ell}, y_{i,\ell}))$ denote the string of oracle query-answer pairs for oracle gates in layer $i$. For a query-answer string $S$, let $S^{\text{in}}$ denote the string of queries in $S$ and $S^{\text{out}}$ denote the string of answers. Decompose $C^O$ into circuits $C_0, \ldots, C_d$ without oracle gates as follows: $C_0$ maps $x \mapsto Q_1^{\text{in}}$; for $i \in [d-1]$, $C_i$ maps $(x, Q_1, \ldots, Q_i) \mapsto Q_{i+1}^{\text{in}}$; and $C_d$ maps $(x, Q_1, \ldots, Q_d) \mapsto C^O(x)$.

**The protocol:**

1. The prover claims $C^O(x) = 1$. The verifier samples a random key $\mathsf{k}$ for a universal hash function family $H$ and sends $\mathsf{k}$ to the prover. For each $i \in [d]$, the prover hashes $Q_i$ under $\mathsf{k}$ to $h_i$ bits and sends the tuple of hashes $(H_{\mathsf{k}}(Q_i))_{i \in [d]}$ to the verifier.

2. The prover and an IPP verifier $\mathcal{V}_d$ run the message-exchange portion of the IPP with proximity parameter $\varepsilon_1 + \cdots + \varepsilon_d$ on input $(x, Q_1, \ldots, Q_d)$ to show that $C_d(x, Q_1, \ldots Q_d) = 1$ and $Q_1, \ldots, Q_d$ hash consistently with the commitments in step 1, where $(Q_1, \ldots, Q_d)$ is the implicit part of the input. At the end, $\mathcal{V}_d$ outputs sets of indices $J_{d,1}, \ldots, J_{d,d} \subseteq [\ell]$ that it wishes to query in $Q_1, \ldots, Q_d$, respectively.

3. The verifier asks the prover to send the query-answer pairs in $Q_i$ at indices $J_{d,i}$ for each $i \in [d]$. For each $i$, the prover sends a string $A_{d,i} := Q_i|_{J_{d,i}}$, and the verifier checks that all the oracle calls in every $A_{d,i}$ are correct and provides $A_{d,1}, \ldots A_{d,d}$ as the answers to $\mathcal{V}_d$'s queries.

4. For $j = d-1, d-2, \ldots, 1$, the prover and an IPP verifier $\mathcal{V}_i$ run an IPP with proximity parameter $\varepsilon_1 + \cdots + \varepsilon_j$ on input $(x, J_{j+1}, A_{j+1}^{\text{in}}, Q_1, \ldots, Q_j)$ to show that $C_j(x, Q_1, \ldots, Q_j)|_{J_{j+1}} = A_{j+1}^{\text{in}}$ and $Q_1, \ldots, Q_j$ hash consistently with the commitments in step 1, where $J_{j+1} = \bigcup_{k=j+1}^{d} J_{k,j+1}$ consists of all the indices that IPP verifiers in previous steps have queried in $Q_{j+1}$, $A_{j+1}$ is the sorted concatenation of $A_{d,j+1}, A_{d-1,j+1}, \ldots, A_{j+1,j+1}$, i.e., all the strings of answers that the prover provides to queries to $Q_{j+1}$ in previous IPPs, and $(Q_1, \ldots, Q_j)$ is the implicit part of the input. At the end, $\mathcal{V}_i$ outputs sets $J_{j,1}, \ldots, J_{j,j} \subseteq [\ell]$ of indices that it wishes to query in $Q_1, \ldots, Q_j$ respectively. For each $i \in [j]$, the verifier asks the prover to send $A_{j,i} := Q_i|_{J_{j,i}}$, checks that the oracle calls in $A_{j,i}$ are correct, and sends $A_{j,i}$ to $\mathcal{V}_i$.

5. Finally, letting $J_1$ denote $\bigcup_{k=1}^{d} J_{k,1}$ and $A_1$ denote the sorted concatenation of $A_{d,1}, \ldots, A_{1,1}$, the prover and a DEIP verifier engage in a DEIP to prove that $C_0(x)|_{J_1} = A_1^{\text{in}}$.

6. The verifier accepts if all of the IPP verifiers accept, the DEIP verifier accepts, and all of its checks pass, and rejects otherwise.

*Figure 2.* DEIP for robust circuits making adaptive queries

We first define some helpful notations. The protocol contains $d$ IPPs, one for each layer of oracle gates in the circuit. We refer to the IPP on implicit input $(Q_1, \ldots, Q_i)$ as IPP $i$ and denote the IPP verifier for IPP $i$ by $\mathcal{V}_i$. The IPPs in our protocol differ from standard IPPs in that the verifier does not have query access to the input: rather, the verifier asks the cheating prover $\widetilde{\mathcal{P}}$ for positions in the input, and $\widetilde{\mathcal{P}}$'s answers may not be consistent with the input. We use $\widetilde{A}_i$ for $\widetilde{\mathcal{P}}$'s string of answers $A_i$ to queries to $Q_i$. We use $\delta_{\text{IPP}}$ and $\delta_{\text{DEIP}}$ for the soundness errors of the IPP and the DEIP, respectively, and set $\delta_{\text{IPP}}, \delta_{\text{DEIP}} \leq 1/d^{10}$ using $O(\log d)$ parallel repetitions. We use $\delta_{\text{hash}}$ for the security parameter of the universal hash function and set $\delta_{\text{hash}} \leq O(1/d)$.

We recursively define strings $R_1, \ldots, R_d$ as follows. Set $R_1 = Q_1$. For $i = 2, \ldots, d$, if there exists a tuple $(R'_1, \ldots, R'_{i-1})$ such that (i) $(R'_1, \ldots, R'_{i-1})$ is $\sum_{j=1}^{i-1} \varepsilon_j$-close to $(R_1, \ldots, R_{i-1})$, (ii) each $R'_j$ is consistent with the hashes that the prover sent in Step 1, and (iii) all oracle calls in each $R'_j$ are correct, then define

$$R_i := \big(C_{i-1}(x, R'_1, \ldots, R'_{i-1}), O(C_{i-1}(x, R'_1, \ldots, R'_{i-1}))\big),$$

where $O(C_{i-1}(x, R'_1, \ldots, R'_{i-1}))$ is obtained by applying $O$ to every query in $C_{i-1}(x, R'_1, \ldots, R'_{i-1})$. We will show that $\widetilde{\mathcal{P}}$'s answers $\widetilde{A}_1, \ldots, \widetilde{A}_d$ are with good probability consistent with $R_1, \ldots, R_d$, which means that even though our IPPs differ from standard IPPs in that the prover holds the implicit input, from $\mathcal{V}_i$'s point of view, IPP $i$ looks the same as standard IPP on implicit input $(R_1, \ldots, R_i)$.

Towards this, we define some helpful events:

- For $i \in [d]$, let $F_i$ be the event that $\widetilde{A}_i$ is consistent with $R_i$.

- For $i \in [d]$, let $H_i$ be the event that there is at most one query-answer string $R'_i$ such that $R'_i$ is $\sum_{i=1}^d \varepsilon_i$-Hamming-close to $R_i$, $R'_i$ hashes consistently with the hashes that the prover sent in step 1, and all the oracle calls in $R'_i$ are correct.

First, we have $\Pr[\overline{F_1}] \leq \delta_{\text{DEIP}}$ since $R_1 = Q_1$, the DEIP verifies that the prover's claimed queries to the first layer are equal to $C_0(x)$ on the queried coordinates, and the verifier also checks the corresponding oracle answers directly. Next, we show that for every $i \in \{2, \ldots, d\}$,

$$\Pr[\overline{F_i} \mid F_1, \ldots, F_{i-1}, H_1, \ldots, H_{i-1}, \mathcal{V} \text{ accepts}] \leq \delta_{\text{IPP}}.$$

Conditioned on $F_1, \ldots, F_{i-1}$, $\mathcal{V}_{i-1}$ sees answers exactly as if it had query access to the implicit input $(R_1, \ldots, R_{i-1})$. Since the outer verifier accepts, $\mathcal{V}_{i-1}$ accepts. Thus, by IPP soundness, except with probability $\delta_{\text{IPP}}$, there exists a tuple $(R'_1, \ldots, R'_{i-1})$ that is $\sum_{j=1}^{i-1} \varepsilon_j$-close to $(R_1, \ldots, R_{i-1})$, hashes consistently with the hashes in step 1, has correct oracle answers, and satisfies the relation checked by IPP $i - 1$. By $H_1, \ldots, H_{i-1}$, each such $R'_j$ is unique, which means that $R_i = (C_{i-1}(x, R'_1, \ldots, R'_{i-1}), O(C_{i-1}(x, R'_1, \ldots, R'_{i-1})))$. Since $\widetilde{A}_i^{\text{in}} = C_{i-1}(x, R'_1, \ldots, R'_{i-1})|_{J_i}$ and the verifier checks that all the oracle calls in $\widetilde{A}_i$ are correct, $\widetilde{A}_i$ is consistent with $R_i$.

Now condition on $F_1, \ldots, F_d$ and $H_1, \ldots, H_d$. Then the final IPP verifier $\mathcal{V}_d$ has exactly the view of a standard IPP verifier with query access to implicit input $(R_1, \ldots, R_d)$. Since the outer verifier accepts, $\mathcal{V}_d$ accepts. Therefore, except with probability $\delta_{\text{IPP}}$, there exists a tuple $(R'_1, \ldots, R'_d)$ that is $\sum_{j=1}^d \varepsilon_j$-close to $(R_1, \ldots, R_d)$, hashes consistently with the hashes in step 1, has correct oracle answers, and satisfies

$$C_d(x, R'_1, \ldots, R'_d) = 1.$$

By $H_1, \ldots, H_d$, this tuple is unique.

Then $(R'_1, \ldots, R'_d)$ is $\sum_{j=1}^d \varepsilon_j$-close to the true query-answer string $(Q_1, \ldots, Q_d)$ in the sense of Definition 4.1: starting from $R_1 = Q_1$, we can modify positions to obtain $R'_1$, then recompute the queries and answers for the next layer from $(x, R'_1)$ and modify positions to obtain $R'_2$, and continue in this way through layer $d$. In total, we modify at most an $\sum_{i=1}^d \varepsilon_i$-fraction of positions. Since $C$ is $\varepsilon$-robust and $C_d(x, R'_1, \ldots, R'_d) = 1$, assuming $\sum_{i=1}^d \varepsilon_i \leq \varepsilon$, we have that $C^O(x) = 1$, which is a contradiction.

Thus, if $x \notin L$ and the verifier accepts, at least one of the following bad events must occur:

$$\overline{F_1}, \quad \overline{H_i} \text{ for some } i \in [d], \quad \overline{F_i} \text{ for some } i \geq 2, \quad \text{or soundness fails in IPP } d.$$

This implies that

$$\Pr[\mathcal{V} \text{ accepts}] \leq \delta_{\mathrm{DEIP}} + d\delta_{\mathrm{hash}} + (d-1)\delta_{\mathrm{IPP}} + \delta_{\mathrm{IPP}} = \delta_{\mathrm{DEIP}} + d\delta_{\mathrm{hash}} + d\delta_{\mathrm{IPP}},$$

which, substituting, is at most $p$. It remains to justify the bound $\delta_{\mathrm{hash}}$ on $\Pr[\overline{H_i}]$. For any $i$, the number of strings that are $\sum_{j=1}^{d} \varepsilon_j$-Hamming-close to $R_i$ is at most

$$N_i \leq \ell^{\ell \sum_{j=1}^{d} \varepsilon_j}.$$

For any alternative candidate string $S \neq R_i$, the probability over the hash key that all blocks of $S$ hash consistently with the committed hashes is at most $2^{-h_i}$. So by a union bound, $\Pr[\overline{H_i}] \leq N_i \cdot 2^{-h_i}$. Choosing

$$h_i \geq \ell\Big(\sum_{j=1}^{d} \varepsilon_j\Big) \log \ell + \log(1/\delta_{\mathrm{hash}})$$

makes this at most $\delta_{\mathrm{hash}}$.

**Efficiency.** Finally, we set the parameters and calculate the efficiency of the protocol. We take

- $\varepsilon_i = \varepsilon/d$ for all $i \in [d]$, and

- $h_i = \varepsilon\ell \log \ell + \log(1/\delta_{\mathrm{hash}}) = O(\varepsilon\ell \log \ell + \log d)$.

To analyze the efficiency, we decompose the protocol into the following parts:

- The prover sends all hashes after getting the hash key. This involves $O(1)$ rounds and $\sum_i h_i = \mathsf{poly}(d) \cdot \varepsilon\mathsf{polylog}(S)$ communication.

- The prover and verifier engage in $d$ IPPs, one for each layer of oracle gates. This involves $O(dD \cdot \mathsf{polylog}(S))$ rounds, $\sum_i \varepsilon_i\ell \cdot D\mathsf{polylog}(S) = \varepsilon DS\mathsf{polylog}(S)$ communication, and $O(\log d \cdot \sum_i 1/\varepsilon_i) = \mathsf{poly}(d)/\varepsilon$ oracle queries. The running time of each IPP verifier is $\varepsilon_i(\sum_i h_i) \cdot (\sum_i 1/\varepsilon_i) \cdot \mathsf{poly}(D, \log S) + \mathsf{polylog}(S)/\varepsilon_i = \varepsilon S\mathsf{poly}(d, D, \log S) + \mathsf{poly}(d, \log S)/\varepsilon$, so the total verification time is $\varepsilon S\mathsf{poly}(d, D, \log S) + \mathsf{poly}(d, \log S)/\varepsilon$.

- The prover and verifier engage in a DEIP. This involves $D\mathsf{polylog}(S)$ rounds, $D\mathsf{polylog}(S)$ communication, and $\mathsf{poly}(d, D, \log S)/\varepsilon$ verification time.

Therefore, the protocol satisfies the following parameters:

- soundness error $1/2$;

- query complexity $q = \mathsf{poly}(d)/\varepsilon$;

- communication complexity $cc = \varepsilon DS\mathsf{polylog}(S)$;

- round complexity $dD\mathsf{polylog}(S)$;

- verifier running time $\mathsf{poly}(d, D, \log S)(\varepsilon S + 1/\varepsilon)$; and

- prover running time $\mathsf{poly}(S)$.

$\square$

## C.3. An argument system for circuits making adaptive oracle queries

*Theorem* C.3. Let $\varepsilon = \varepsilon(n) \in (0, 1]$ be a proximity parameter, and let $\mathcal{L}$ be a language that has $\varepsilon$-robust oracle Boolean circuits with $\ell = \ell(n)$ oracle gates and of polynomial size $S = S(n) \leq 2^\lambda$. Then there is a doubly-efficient interactive argument system for $\mathcal{L}$ with the following parameters:

- soundness error $1/2$;

- query complexity $q = O(1/\varepsilon)$;

- communication complexity $cc = \mathsf{poly}(\lambda) \cdot 1/\varepsilon$;

- round complexity $O(1)$;

- verifier running time $\mathsf{poly}(\lambda) \cdot (1/\varepsilon + n)$; and

- prover running time $\mathsf{poly}(S) \cdot 1/\varepsilon$.

---

**Doubly-efficient interactive argument for robust, adaptive setting**

**Input:** The prover $\mathcal{P}$ and verifier $\mathcal{V}$ receive $x \in \{0,1\}^n$, a succinct description of a logspace-uniform oracle circuit $C$ containing $\ell$ oracle gates, and oracle access to $O : \{0,1\}^* \to \{0,1\}$.

**Notations and tools:** Enumerate the oracle gates of $C^O$ in topological order by $1, \ldots, \ell$. On input $x$, for each $i \in [\ell]$, let $x_i$ denote the input to the $i$th oracle gate and $y_i = O(x_i)$ the corresponding oracle answer. Let $\mathcal{M}$ denote the set of oracle input-output pairs. We will use the following tools:

- A succinct argument of knowledge $\mathsf{KIL} = (\mathsf{Setup}_{\mathsf{Kil}}, \mathcal{P}_{\mathsf{Kil}}, \mathcal{V}_{\mathsf{Kil}}, \mathsf{Ext}_{\mathsf{Kil}})$ for NP relations (see Section B.4).

- A hash tree family $\mathsf{HT} = (\mathcal{G}_{\mathsf{ht}}, \mathcal{H}_{\mathsf{ht}}, \mathcal{P}_{\mathsf{ht}}, \mathcal{V}_{\mathsf{ht}})$ (see Definition B.8).

**The protocol:**

1. The verifier generates public parameters $\mathsf{pp}_{\mathsf{Kil}} \leftarrow \mathsf{Setup}_{\mathsf{Kil}}(1^\lambda)$, and $\mathsf{hk} \leftarrow \mathcal{G}_{\mathsf{ht}}(1^\lambda, \ell)$, where $\mathsf{hk}$ are hash keys for database over $\mathcal{M}$.

2. The prover simulates the entire execution $C^O(x) = 1$, computing all intermediate oracle queries $x_1, \ldots, x_\ell$ and answers $y_1 = O(x_1), \ldots, y_\ell = O(x_\ell)$. The prover then computes the hash tree root $\mathsf{rt} \leftarrow \mathcal{H}_{\mathsf{ht}}(\mathsf{hk}, \{(x_1, y_1), \ldots, (x_\ell, y_\ell)\})$ and sends $\mathsf{rt}$ to the verifier.

3. The prover and verifier engage in Kilian's protocol to prove there that there exists a set $\{(x_1^*, y_1^*), \ldots, (x_\ell^*, y_\ell^*)\}$ that is (i) consistent with $\mathsf{rt}$ and (ii) on input $x$, taking the oracle answers to be those in $\{y_i^*\}_{i \in [\ell]}$, $C$ makes the queries given in $\{x_i^*\}_{i \in [\ell]}$ and accepts. In particular, they prove the NP relation

$$\mathcal{R} := \{(\mathsf{hk}, (C, x, \mathsf{rt}), \{(x_1^*, y_1^*), \ldots, (x_\ell^*, y_\ell^*)\}) \mid \mathsf{rt} = \mathcal{H}_{\mathsf{ht}}(\mathsf{hk}, \{(x_1^*, y_1^*), \ldots, (x_\ell^*, y_\ell^*)\}) \wedge C^{x_i^* \to y_i^*}(x) = 1\}.$$

4. The verifier randomly samples $q$ indices $J \subseteq [\ell]$ and asks the prover to open the hash tree on these indices. The prover sends $\{(x_j^*, y_j^*, \rho_j)\}_{j \in J}$, where $\rho_j$ is the hash tree opening proof for $(x_j^*, y_j^*)$. The verifier checks that for all $j \in J$, $\mathcal{V}_{\mathsf{ht}}(\mathsf{hk}, \mathsf{rt}, (x_j^*, y_j^*), \rho_j) = 1$ and $y_j^* = O(x_j^*)$.

5. The verifier accepts if all checks pass, and rejects otherwise.

*Figure 3.* Doubly-efficient interactive argument for robust circuits making adaptive queries

---

*Proof.* The protocol appears in Figure 3. Here we analyze its efficiency, completeness, and soundness. Set $q = 2/\varepsilon$.

**Efficiency.** The communication is composed of:

- Public parameters $\mathsf{pp}_{\mathsf{Kil}}, \mathsf{hk}$ of size $\mathsf{poly}(\lambda)$.

- Hash root $\mathsf{rt}$ of size $\mathsf{poly}(\lambda)$.

- The Kilian's protocol interaction of size $\mathsf{poly}(\lambda)$.

- $q$ oracle input-output pairs and their openings $\left\{(x_j^*, y_j^*, \rho_j)\right\}_{j \in J}$, where $q = O(1/\varepsilon)$. Their total size is $\mathsf{poly}(\lambda) \cdot 1/\varepsilon$.

Thus the total communication is $\mathsf{poly}(\lambda) \cdot 1/\varepsilon$.

During the above protocol, the verifier runs in time $\mathsf{poly}(\lambda) \cdot (1/\varepsilon + n)$ and the prover runs in time $\mathsf{poly}(S) \cdot 1/\varepsilon$.

**Completeness.** The completeness follows from the completeness of Kilian's protocol and the hash tree.

**Soundness.** Let $\mathcal{P}^*$ be any poly-time malicious prover, and we rewind $\mathcal{P}^*$ (just after sending $\mathsf{rt}$) to obtain a prover for the succinct argument of knowledge $\mathcal{P}_1^*$. Specifically, we run $\mathsf{Ext}_{\mathsf{Kil}}^{\mathcal{P}_1^*}$ to extract $\{(x_1^*, y_1^*), \ldots, (x_\ell^*, y_\ell^*)\}$ that is consistent with $\mathsf{rt}$ and the computation. By knowledge soundness, we have

$$
\Pr\left[
\begin{array}{c}
\langle \mathcal{P}^*, \mathcal{V} \rangle = 1 \wedge \\
(\mathsf{rt} \neq \mathcal{H}_{\mathsf{ht}}(\mathsf{hk}, \{(x_1^*, y_1^*), \ldots, (x_\ell^*, y_\ell^*)\}) \\
\vee C^{x_i^* \to y_i^*}(x) = 0)
\end{array}
:
\begin{array}{c}
\mathsf{pp}_{\mathsf{Kil}} \leftarrow \mathsf{Setup}_{\mathsf{Kil}}(1^\lambda), \mathsf{hk} \leftarrow \mathcal{G}_{\mathsf{ht}}(1^\lambda, \ell) \\
\{(x_1^*, y_1^*), \ldots, (x_\ell^*, y_\ell^*)\} \leftarrow \mathsf{Ext}_{\mathsf{Kil}}^{\mathcal{P}_1^*}
\end{array}
\right] \leq \mathsf{negl}(\lambda).
$$

Now we claim that for at most an $\varepsilon$ fraction of $\{(x_1^*, y_1^*), \ldots, (x_\ell^*, y_\ell^*)\}$, it holds that $y_i^* \neq O(x_i^*)$.

*Claim* C.1. For any poly-time malicious prover $\mathcal{P}^*$, it holds that

$$
\Pr\left[
\begin{array}{c}
\langle \mathcal{P}^*, \mathcal{V} \rangle = 1 \\
\wedge \mathsf{rt} = \mathcal{H}_{\mathsf{ht}}(\mathsf{hk}, \{(x_1^*, y_1^*), \ldots, (x_\ell^*, y_\ell^*)\}) \\
\wedge \big| \{i \in [\ell] : y_i^* \neq O(x_i^*)\} \big| > \varepsilon \cdot \ell
\end{array}
:
\begin{array}{c}
\mathsf{pp}_{\mathsf{Kil}} \leftarrow \mathsf{Setup}_{\mathsf{Kil}}(1^\lambda), \mathsf{hk} \leftarrow \mathcal{G}_{\mathsf{ht}}(1^\lambda, \ell) \\
\{(x_1^*, y_1^*), \ldots, (x_\ell^*, y_\ell^*)\} \leftarrow \mathsf{Ext}_{\mathsf{Kil}}^{\mathcal{P}_1^*}
\end{array}
\right] \leq \frac{1}{3}.
$$

*Proof.* We consider two events:

- $E_0$: the sampled $q$ indices $J$ contains at least one index $j$ such that $y_j^* \neq O(x_j^*)$.

- $E_1$: the sampled $q$ indices $J$ contains no index $j$ such that $y_j^* \neq O(x_j^*)$.

If event $E_0$ happens and the verifier accepts, then $\mathcal{P}^*$ must break the binding property of the hash tree. That is,

$$
\Pr\left[
\begin{array}{c}
\langle \mathcal{P}^*, \mathcal{V} \rangle = 1 \\
\wedge \mathsf{rt} = \mathcal{H}_{\mathsf{ht}}(\mathsf{hk}, \{(x_1^*, y_1^*), \ldots, (x_\ell^*, y_\ell^*)\}) \\
\wedge \big| \{i \in [\ell] : y_i^* \neq O(x_i^*)\} \big| > \varepsilon \cdot \ell \\
\wedge E_0
\end{array}
:
\begin{array}{c}
\mathsf{pp}_{\mathsf{Kil}} \leftarrow \mathsf{Setup}_{\mathsf{Kil}}(1^\lambda), \mathsf{hk} \leftarrow \mathcal{G}_{\mathsf{ht}}(1^\lambda, \ell) \\
\{(x_1^*, y_1^*), \ldots, (x_\ell^*, y_\ell^*)\} \leftarrow \mathsf{Ext}_{\mathsf{Kil}}^{\mathcal{P}_1^*}
\end{array}
\right] \leq \mathsf{negl}(\lambda).
$$

On the other hand, the probability for $E_1$ to happen is small. In particular, $\Pr[E_1] \leq (1 - \varepsilon)^q$. Since we set $q = \frac{2}{\varepsilon}$, we have $\Pr[E_1] \leq 1/e^2$.

Combining the two cases we have

$$
\Pr\left[
\begin{array}{c}
\langle \mathcal{P}^*, \mathcal{V} \rangle = 1 \\
\wedge \mathsf{rt} = \mathcal{H}_{\mathsf{ht}}(\mathsf{hk}, \{(x_1^*, y_1^*), \ldots, (x_\ell^*, y_\ell^*)\}) \\
\wedge \big| \{i \in [\ell] : y_i^* \neq O(x_i^*)\} \big| > \varepsilon \cdot \ell
\end{array}
:
\begin{array}{c}
\mathsf{pp}_{\mathsf{Kil}} \leftarrow \mathsf{Setup}_{\mathsf{Kil}}(1^\lambda), \mathsf{hk} \leftarrow \mathcal{G}_{\mathsf{ht}}(1^\lambda, \ell) \\
\{(x_1^*, y_1^*), \ldots, (x_\ell^*, y_\ell^*)\} \leftarrow \mathsf{Ext}_{\mathsf{Kil}}^{\mathcal{P}_1^*}
\end{array}
\right] \leq \mathsf{negl}(\lambda) + \frac{1}{e^2} \leq \frac{1}{3}.
$$

$\square$

Therefore, if $\mathcal{P}^*$ convinces the verifier, then it must be the case that $C^O(x)$ accepts with at most an $\varepsilon$ fraction of incorrect oracle answers, that is, with probability at most $1/3 + \mathsf{negl}(\lambda)$, we have $C^{x_i^* \to y_i^*}(x) = 1$ and $\big| \{i \in [\ell] : y_i^* \neq O(x_i^*)\} \big| \leq \varepsilon \cdot \ell$. This finishes our soundness analysis. $\square$

## D. A Doubly-Efficient Interactive Argument for Low-Degree Oracles

Here we present our doubly-efficient argument system for circuits with access to a low-degree oracle.

*Theorem* D.1. Let $O : \{0,1\}^* \to \{0,1\}^*$ be an oracle with degree $d(n) = \mathsf{poly}(n)$, $\varepsilon = \varepsilon(n) \in (0,1]$ be a proximity parameter, and let $\mathcal{L}$ be a language that has $\varepsilon$-robust logspace-uniform oracle Boolean circuits with $S = \mathsf{poly}(n)$ gates. Take $\lambda$ as the security parameter. Assuming the polynomial hardness of LWE, there exists a interactive argument system for $\mathcal{L}$ with the following parameters:

- query complexity $q = 1$;

- communication complexity $cc = \mathsf{poly}(\lambda)$;

- round complexity $O(\log n)$;

- verifier running time $\mathsf{poly}(\lambda) \cdot n$; and

- prover running time $\mathsf{poly}(\lambda, S)$.

*Proof.* The protocol appears in Figure 4. We analyze its efficiency, completeness, and soundness as follows.

**Efficiency.** The communication of this protocol is composed of:

- Public parameters pp of size $\mathsf{poly}(\lambda)$.

- Commitments $(\mathsf{c}_G, \mathsf{c}_F)$ of size $\mathsf{poly}(\lambda)$.

- Kilian's protocol communication, of size $\mathsf{poly}(\lambda)$.

- The challenge $u$ of size $\log |\mathbb{F}| = O(1)$.

- The openings $(v_G, v_F)$ of size $O(1)$.

- The polynomial commitment opening, of size $\mathsf{poly}(\lambda)$.

Therefore, the total communication is $\mathsf{poly}(\lambda)$.

For round complexity, since the Kilian's protocol has $O(1)$ rounds, and PC local opening argument has $O(\log(S \cdot d(m))) = O(\log n)$ rounds, the total number of rounds is $O(\log n)$.

The computation of the verifier is composed of Kilian's protocol verifier, polynomial commitment verifier and checking $v_F = O(v_G)$. Therefore the running time is $\mathsf{poly}(\lambda) \cdot n$, and the query complexity is 1.

**Completeness.** Completeness follows from completeness of the polynomial commitment scheme and Kilian's protocol.

**Soundness.** Let $\mathcal{P}^*$ be any malicious prover. To prove computational soundness of this protocol, we rewind $\mathcal{P}^*$ to extract the witness during the invocation of Kilian's protocol. Specifically, we run $\mathsf{Ext}_{\mathsf{Kil}}^{\mathcal{P}_2^*}$ to extract $(G^*, \mathsf{st}_G^*, F^*, \mathsf{st}_F^*)$.

Since the protocol also performs Kilian's protocol verification, by definition of knowledge extration, we know that if $\mathcal{P}^*$ convinces the verifier, then $G^*, F^*$ extracted by $\mathsf{Ext}_{\mathsf{Kil}}^{\mathcal{P}_2^*}$ must be valid, in the sense that they are the polynomials underlying $\mathsf{c}_G, \mathsf{c}_F$, and $C(x)$ accepts when making queries by $G^*$ and using oracle answers from $F^*$, which we denote by $C^{G^* \to F^*}(x) = 1$.

*Claim* D.1. For any poly-time prover $\mathcal{P}^*$, there exists a negligible function $\mathsf{negl}(\cdot)$ such that

$$\Pr \left[ \begin{array}{c} \langle \mathcal{P}^*, \mathcal{V} \rangle(\mathsf{pp}, C, x) = 1 \wedge \\ \left( \mathsf{Open}(\mathsf{pp}_{\mathsf{CM}}, \mathsf{c}_G, G^*, \mathsf{st}_G^*) = 0 \right. \\ \vee \mathsf{Open}(\mathsf{pp}_{\mathsf{CM}}, \mathsf{c}_F, F^*, \mathsf{st}_F^*) = 0 \\ \left. \vee C^{G^* \to F^*}(x) = 0 \right) \end{array} : \begin{array}{c} \mathsf{pp} \leftarrow \mathsf{Setup}(1^\lambda, S) \\ (G^*, \mathsf{st}_G^*, F^*, \mathsf{st}_F^*) \leftarrow \mathsf{Ext}_{\mathsf{Kil}}^{\mathcal{P}_2^*} \end{array} \right] \le \mathsf{negl}(\lambda).$$

---

**Doubly-efficient interactive argument for circuits with access to a low-degree oracle**

**Input:** The prover $\mathcal{P}$ and verifier $\mathcal{V}$ receive $x \in \{0,1\}^n$, a succinct description of an oracle circuit $C$, and oracle access to $O : \{0,1\}^* \rightarrow \{0,1\}$ with degree $d(n) = \mathsf{poly}(n)$.

**Notations and tools:** Take $m = m(\lambda)$ to be the maximum input length to the oracle. We take the field $\mathbb{F} = \mathbb{F}_\lambda$ on which $O_m$ is represented as a poly-degree polynomial. We assume that $|\mathbb{F}| = 2^{O(m)}$. Let $S$ denote the size of $C$. Our construction uses the following building blocks:

- A polynomial commitment scheme PC = $(\mathsf{Setup_{CM}}, \mathsf{Commit}, \mathsf{Open}, \mathsf{Setup_{IP}}, \mathcal{P}_{\mathsf{PC}}, \mathcal{V}_{\mathsf{PC}})$ with efficient opening and verification (e.g., Theorem B.4), for degree $S \cdot d(m)$ polynomials over a finite field $\mathbb{F}$.

- A succinct argument of knowledge KIL = $(\mathsf{Setup_{Kil}}, \mathcal{P}_{\mathsf{Kil}}, \mathcal{V}_{\mathsf{Kil}}, \mathsf{Ext_{Kil}})$ for NP relations (see Section B.4).

We use $\mathcal{P}_1, \mathcal{V}_1, \ldots$ to describe different phases of interaction between the two parties. Note that $\mathcal{P}_i, \mathcal{V}_i$ does not necessarily mean $\mathcal{P}, \mathcal{V}$ in the $i$-th round. We assume both $\mathcal{P}$ and $\mathcal{V}$ keep their state.

**The protocol:**

1. $\mathsf{pp} \leftarrow \mathsf{Setup}(1^\lambda, S)$: Sample $\mathsf{pp_{CM}} \leftarrow \mathsf{Setup_{CM}}(1^\lambda, S \cdot d(m))$, $\mathsf{pp_{IP}} \leftarrow \mathsf{Setup_{IP}}(1^\lambda)$, and $\mathsf{pp_{Kil}} \leftarrow \mathsf{Setup_{Kil}}(1^\lambda)$, and output $\mathsf{pp} = (\mathsf{pp_{CM}}, \mathsf{pp_{IP}}, \mathsf{pp_{Kil}})$.

2. $(\mathsf{c}_G, \mathsf{c}_F) = \mathcal{P}_1^O(C, x, \mathsf{pp})$:
   - Simulate the computation $C^O(x) = 1$, writing down the oracle queries $x_1, \ldots, x_S$ made and their corresponding oracle answers $y_1 = O(x_1), \ldots, y_S = O(x_S)$.
   - Construct polynomial $G$ of degree $S$ such that $G(i) = x_i$ for every $i \in [S]$. Construct polynomial $F$ as the unique polynomial of degree $S \cdot d(m)$ such that $F(x) = O(G(x))$ for every $x \in \mathbb{F}$. Note that $F$ can be computed by at most $S \cdot d(m)$ queries to the oracle.
   - Compute $(\mathsf{c}_G, \mathsf{st}_G) \leftarrow \mathsf{Commit}(\mathsf{pp_{CM}}, G)$ and $(\mathsf{c}_F, \mathsf{st}_F) \leftarrow \mathsf{Commit}(\mathsf{pp_{CM}}, F)$.
   - Send $(\mathsf{c}_G, \mathsf{c}_F)$ to the verifier.

3. $\langle \mathcal{P}_2, \mathcal{V}_1 \rangle$ simulate the interaction $\langle \mathcal{P}_{\mathsf{Kil}}, \mathcal{V}_{\mathsf{Kil}} \rangle$ for the relation $\mathcal{R}$ defined as $(\mathsf{pp_{CM}}, (C, x, \mathsf{c}_G, \mathsf{c}_F), (G, \mathsf{st}_G, F, \mathsf{st}_F)) \in \mathcal{R}$ if and only if, if we run the computation $C$ on $x$, but with the the oracle answers $y_i$ replaced by $F(i)$, all oracle queries are correct corresponding to $G$, the computation accepts, and $\mathsf{Open}(\mathsf{pp_{CM}}, \mathsf{c}_G, G, \mathsf{st}_G) = \mathsf{Open}(\mathsf{pp_{CM}}, \mathsf{c}_F, F, \mathsf{st}_F) = 1$. $\mathcal{V}_1$ rejects if $\mathcal{V}_{\mathsf{Kil}}$ rejects.

4. $u \leftarrow \mathcal{V}_2$: The verifier samples $u \leftarrow \mathbb{F}$ and sends $u$ to the prover.

5. $(v_G, v_F) = \mathcal{P}_3$: The prover computes $v_G = G(u)$ and $v_F = F(u)$ and sends $(v_G, v_F)$ to the verifier.

6. $\langle \mathcal{P}_4, \mathcal{V}_3 \rangle$ simulate the interaction $\langle \mathcal{P}_{\mathsf{PC}}, \mathcal{V}_{\mathsf{PC}} \rangle(\mathsf{pp_{IP}})$ for the opening proof of $v_G$ and $v_F$. $\mathcal{V}_3$ rejects if $\mathcal{V}_{\mathsf{PC}}$ rejects.

7. $b \leftarrow \mathcal{V}_4$: The verifier accepts if $\mathcal{V}_1$ and $\mathcal{V}_3$ accept and $v_F = O(v_G)$ and rejects otherwise.

*Figure 4.* Doubly-efficient interactive argument for circuits with access to a low-degree oracle

Then we claim that our protocol ensures $F^* = O(G^*)$ with all but negligible probability.

*Claim* D.2. For any poly-time prover $\mathcal{P}^*$, there exists a negligible function $\mathsf{negl}(\cdot)$ such that

$$\Pr\left[\ \langle\mathcal{P}^*,\mathcal{V}\rangle(\mathsf{pp},C,x) = 1 \wedge F^* \neq O(G^*)\ :\ \begin{array}{c}\mathsf{pp} \leftarrow \mathsf{Setup}(1^\lambda, S) \\ (G^*,\mathsf{st}_G^*,F^*,\mathsf{st}_F^*) \leftarrow \mathsf{Ext}_{\mathsf{Kil}}^{\mathcal{P}_2^*}\end{array}\ \right] \leq \mathsf{negl}(\lambda).$$

*Proof.* We bound the probability by considering whether $F^*(u) = O(G^*(u))$.

First, since both $F^*$ and $O(G^*(u))$ are poly-degree polynomials over $\mathbb{F}$, by Schwartz-Zippel, we know that $F^*$ and $O(G^*)$ can agree on at most $\deg(F^*) \leq S \cdot d(m)$ points. Therefore, as $u$ is sampled uniformly from $\mathbb{F}$, we have

$$\Pr\left[\ \begin{array}{c}\langle\mathcal{P}^*,\mathcal{V}\rangle(\mathsf{pp},C,x) = 1 \\ \wedge F^* \neq O(G^*) \wedge F^*(u) = O(G^*(u))\end{array}\ :\ \begin{array}{c}\mathsf{pp} \leftarrow \mathsf{Setup}(1^\lambda, S) \\ (G^*,\mathsf{st}_G^*,F^*,\mathsf{st}_F^*) \leftarrow \mathsf{Ext}_{\mathsf{Kil}}^{\mathcal{P}_2^*}\end{array}\ \right] \leq \frac{S \cdot d(m)}{2^\lambda}.$$

Otherwise, if $F^*(u) \neq O(G^*(u))$, then $\mathcal{P}^*$ must break the polynomial commitment binding property to convince the verifier. In particular, either $v_G \neq G(u)$ or $v_F \neq F(u)$. Therefore, we can construct an adversary that breaks the binding property, by honestly computing $v_G, v_F$ and provide their openings. This gives us that for some $\mathsf{negl}(\cdot)$,

$$\Pr\left[\ \begin{array}{c}\langle\mathcal{P}^*,\mathcal{V}\rangle(\mathsf{pp},C,x) = 1 \\ \wedge F^* \neq O(G^*) \wedge F^*(u) \neq O(G^*(u))\end{array}\ :\ \begin{array}{c}\mathsf{pp} \leftarrow \mathsf{Setup}(1^\lambda, S) \\ (G^*,\mathsf{st}_G^*,F^*,\mathsf{st}_F^*) \leftarrow \mathsf{Ext}_{\mathsf{Kil}}^{\mathcal{P}_2^*}\end{array}\ \right] \leq \mathsf{negl}(\lambda).$$

Combining the two cases, we conclude the proof. □

Now since $C^{G^* \to F^*}(x) = 1$ and $F^* = O(G^*)$, we have that $C^O(x) = 1$. Therefore,

$$\Pr\left[\ \begin{array}{c}\langle\mathcal{P}^*,\mathcal{V}\rangle(\mathsf{pp},C,x) = 1 \wedge \\ C^O(x) = 0\end{array}\ :\ \mathsf{pp} \leftarrow \mathsf{Setup}(1^\lambda, S)\ \right] \leq \mathsf{negl}(\lambda).$$

□

