# OpenReview forum: "How to Avoid Debate: Scalable AI Safety via Doubly-Efficient Interactive Proofs"
_ICML.cc/2026/Conference — ICML 2026 regular_

### Official Review · Reviewer_15ra · 2026-03-02

**Soundness:** 4
**Presentation:** 4
**Significance:** 2
**Originality:** 4
**Overall Recommendation:** 3
**Confidence:** 2

**Summary:**

This paper studies single-prover interactive proofs for oracle-aided computation as a foundation for scalable AI safety. Prior work has relied on debate-based verification, which requires two equally capable AI systems and assumes the existence of a truthful debater. The authors show that debate is not strictly necessary: although general oracle-aided computation does not admit relativizing interactive proofs, doubly-efficient single-prover protocols can be constructed under two natural structural assumptions, the computation is robust to a small fraction of incorrect oracle answers, or the oracle can be represented by a low-degree polynomial. Under these settings, the verifier runs in near-linear time and makes only a small (or even constant) number of oracle queries. The paper positions these results as a theoretical step toward scalable verification without debate and outlines future directions including improving efficiency, extending to broader classes of learnable oracles, formalizing robustness assumptions, and studying settings where the verifier lacks even a full description of the task.

**Compliance With Llm Reviewing Policy:**

Affirmed.

**Final Justification:**

Thank you for the clarification. The rebuttal provides helpful intuition, especially regarding robustness. However, my main concern is about how broadly the structural assumptions apply to realistic AI safety tasks, which remains unclear. I also understand the conceptual positioning of the work, but the connection to practical ML systems is still limited. These concerns are unlikely to be resolved within the scope of a short rebuttal. After considering these aspects, I decide to keep my current rating.

**Key Questions For Authors:**

No questions.

**Limitations:**

Yes

**Strengths And Weaknesses:**

Strenghts:
- The paper provides a principled single-prover alternative to debate-based scalable oversight, showing that relativizing doubly-efficient verification is possible under structured oracle assumptions.
- The protocols for robust computation and low-degree oracles are carefully constructed using modern cryptographic tools, with clearly stated efficiency guarantees.

Weaknesses:
- The main results depend on fairly strong structural assumptions, namely that the computation is ε-robust or that the oracle can be represented as a low-degree polynomial. While the paper argues that these settings are “natural” and provides some intuition (e.g., robustness via redundancy, or human feedback being structured), it is not entirely clear how often these assumptions hold in realistic AI safety tasks. For example, many reasoning or generation tasks may be sensitive to small oracle errors, and human judgment may not exhibit the strong algebraic structure implied by low-degree models. A more concrete discussion of where these assumptions are expected to hold in practice would strengthen the paper.
- While the theoretical constructions are well-developed, the paper explicitly states that the protocols are not intended for direct practical implementation. The work does not include empirical validation, system-level evaluation, or concrete integration into modern LLM training pipelines. As a result, the practical implications for current AI systems remain unclear, and it would be helpful to better articulate how these theoretical results could eventually translate into deployable scalable oversight mechanisms.

---

> ### Author Rebuttal · Authors · 2026-03-31
>
> Thank you for your comments! We would like to address some of the points you raised.
>
> > The main results depend on fairly strong structural assumptions, namely that the computation is ε-robust or that the oracle can be represented as a low-degree polynomial. While the paper argues that these settings are “natural” and provides some intuition (e.g., robustness via redundancy, or human feedback being structured), it is not entirely clear how often these assumptions hold in realistic AI safety tasks. For example, many reasoning or generation tasks may be sensitive to small oracle errors, and human judgment may not exhibit the strong algebraic structure implied by low-degree models. A more concrete discussion of where these assumptions are expected to hold in practice would strengthen the paper
>
> We believe that, especially in the case when the oracle represents human judgment, it makes sense to assume that the computation is robust. Human judgment is already unstable and error-prone, so as a safeguard, the output of the computation should not change if a small number of the human answers change, otherwise the output is not very meaningful.
>
> > While the theoretical constructions are well-developed, the paper explicitly states that the protocols are not intended for direct practical implementation. The work does not include empirical validation, system-level evaluation, or concrete integration into modern LLM training pipelines. As a result, the practical implications for current AI systems remain unclear, and it would be helpful to better articulate how these theoretical results could eventually translate into deployable scalable oversight mechanisms.
>
> We agree that experimental validation is an important next step. However, we view our contribution as conceptual. We wanted to send a message to the ML community that debate, which previous works have relied on, is actually not necessary, and to present single-prover interactive proofs as an alternative. As such, our contribution is to define a framework of single-prover interactive proofs for scalable oversight and to prove its feasibility via theoretical results. Our hope is that our work will serve as a theoretical foundation for future works that run experiments.

---

> > ### Author Rebuttal · Reviewer_15ra · 2026-04-03
> >
> > Thank you for the clarification. The rebuttal provides helpful intuition, especially regarding robustness. However, my main concern is about how broadly the structural assumptions apply to realistic AI safety tasks, which remains unclear. I also understand the conceptual positioning of the work, but the connection to practical ML systems is still limited. These concerns are unlikely to be resolved within the scope of a short rebuttal.

---

### Official Review · Reviewer_gfNn · 2026-03-11

**Soundness:** 3
**Presentation:** 3
**Significance:** 3
**Originality:** 3
**Overall Recommendation:** 4
**Confidence:** 4

**Summary:**

In this paper, the authors proposed a single-prover interactive proof to overcome the limitations of the "AI Debate" model, which h relies on the unrealistic assumption that two competing AI models are equally capable and that one remains truthful. The authors demonstrate that for either robust computation or low-degree oracles, the output of a powerful AI can be verified efficiently and securely using a minimal number of queries.

**Compliance With Llm Reviewing Policy:**

Affirmed.

**Key Questions For Authors:**

(1)In the article, the single prover lacks an opposing party. if the prover is untrustworthy and consistently outputs misleading reasoning, would these results cause the verifier to make an incorrect judgment.
(2)The majority of this paper focuses on theoretical proofs. Could the authors provide relevant practical engineering case studies or deployment workflows (specifically examples using LLMs to act as the prover and verifier)? This would help readers better understand the content of the paper.
(3)Can the protocols in the paper ensure that the LLM's intentions are aligned with human intentions? If so, can they be applied to RLHF to replace the manual feedback process?
(4)The paper lacks a performance analysis of the four-party protocol, such as the computational and latency overheads introduced by the protocol.

**Limitations:**

yes

**Strengths And Weaknesses:**

Strong points:
+ In the low-degree oracle setting, the verifier (human) requires only 1 oracle query to verify extremely complex computational results, significantly reducing oversight cost.
+ This breaks the established perception that achieving AI safety must rely on debate, and initiates research into single-prover interactive proofs for AI safety

Weak points:
-In section 4.2., the verification efficiency decreases significantly as the number of layers $d$ increases, which limits its performance in ultra-deep reasoning tasks.
-Modeling nuanced human semantic judgment as a low-degree polynomial is a significant simplification. It remains unclear if this model can capture non-linear, context-dependent, or subjective preferences essential for alignment

---

> ### Author Rebuttal · Authors · 2026-03-31
>
> Thank you for your comments! We would like to address some of the points you raised.
>
> > In the article, the single prover lacks an opposing party. if the prover is untrustworthy and consistently outputs misleading reasoning, would these results cause the verifier to make an incorrect judgment
>
> No, this would not cause the verifier to make an incorrect judgment (which is our main point, to introduce single-prover interactive proofs as an alternative to debate!). Our protocols are secure even if the prover is untrustworthy (if the prover is untrustworthy, then the verifier will reject). This is captured by the “soundness” guarantee of our protocols.
>
> > The majority of this paper focuses on theoretical proofs. Could the authors provide relevant practical engineering case studies or deployment workflows (specifically examples using LLMs to act as the prover and verifier)?
>
> We agree that experimental validation is an important next step. However, we view our contribution as conceptual. We wanted to send a message to the ML community that debate, which previous works have relied on, is actually not necessary, and to present single-prover interactive proofs as an alternative. As such, our contribution is to define a framework of single-prover interactive proofs for scalable oversight and to prove its feasibility via theoretical results. Our hope is that our work will serve as a theoretical foundation for future works that run experiments.
>
> Please see the response to the next question for examples using LLMs.
>
> > Can the protocols in the paper ensure that the LLM's intentions are aligned with human intentions? If so, can they be applied to RLHF to replace the manual feedback process?
>
> Yes, the protocols in our paper specifically address scalable oversight (which is an AI alignment problem). In the setting where we are training an AI model to perform some time-consuming task, our protocols give a way to efficiently give feedback on the AI’s behavior in a way that aligns with our intentions. Using our protocols for RLHF is definitely a goal: in the RLHF setting, our protocols would give a way to efficiently reward the actions of the agent. Now instead of computing a reward directly, we could have the agent propose a reward and then engage in an interactive protocol with the agent to verify that this reward aligns with our intentions. This would be much more efficient than having to compute the reward ourselves.
>
> > The paper lacks a performance analysis of the four-party protocol, such as the computational and latency overheads introduced by the protocol.
>
> Our paper doesn’t contain a four-party protocol, so we’re not sure what you’re referring to. Can you elaborate?

---

> > ### Author Rebuttal · Reviewer_gfNn · 2026-04-06
> >
> > Thank you for the detailed rebuttal. I have no further questions at this point.

---

### Official Review · Reviewer_WvV4 · 2026-03-11

**Soundness:** 2
**Presentation:** 3
**Significance:** 3
**Originality:** 3
**Overall Recommendation:** 3
**Confidence:** 4

**Summary:**

A proof is said to relativise if you copy-paste it to all worlds so that it holds for any oracle, it is an informal notion but useful for AI safety because of the problem of scalable oversight. Existing proofs don’t relativize and interactive proof systems like Debate have unrealistic assumptions for AI applications which leads to considering single prover interactive proofs. There aren’t any single prover interactive proofs that relativise for all computations but this paper tries to get close by using oracle-aided computation. They study polynomial time provers and efficient verifiers where both have access to oracles.

**Compliance With Llm Reviewing Policy:**

Affirmed.

**Key Questions For Authors:**

1. Are there plans to incorporate this with foundational models or agentic systems?
2. Is using this for Reinforcement Learning one of the goals?

The main claim is that we can model AI safety scenarios or interactions between AI models and humans as interactive proofs with oracle computations because proof systems like Debate do not suffice for the AI safety setting. This is a great idea and the claim is validated well by the proofs provided in the appendices, which are the only evidence necessary for a theoretical result. But the eventual application is meant to be in real world AI models and there is nothing in the paper that describes or plans how to actually apply this. Some experiments comparing the doubly efficient interactive proofs and doubly efficient interactive arguments with the state of the art in the AI alignment landscape would have been nice, specifically for real world AI safety scenarios like Prompt Injections or Hallucinations.
There are some strong claims about efficiency guarantees and it would be nice to see some experiments trying to confirm these theoretical results to see how this kind of interactive proof or interactive argument performs in real AI safety scenarios like Prompt Injections or Hallucinations. Is the eventual goal to apply this sort of verification on RL frameworks?
It makes sense from a cryptography point of view to assume that the prover is honest but it seems hard to imagine how that would look like in a real world application. A prover is honest if they follow the protocol exactly as defined but it is hard to expect an AI model to do so in practice. There might be a possibility of compounding errors due to this assumption in practice. Even if we do assume that the prover is honest, it would be interesting to see a formalisation with a dishonest prover.

**Limitations:**

No; the paper has not discussed potential negative societal impacts of their work.

**Strengths And Weaknesses:**

Strengths
=================
* Innovative and novel approach to AI safety using theoretical cryptography with strong mathematical guarantees
* Runtime guarantees provided as a result of using robustness and TM with oracles
* Efficiency guarantee for doubly efficient interactive proof with low degree oracle, the verifier proved to be querying the oracle only once.
* Adaptable concept as any database can be converted into a low-degree oracle.

Weaknesses
===================
* No experimentation or validation of the runtime efficiency guarantees.
* No comment or description of how to integrate this with existing foundation models
* The final result on doubly efficient IPs with low degree oracles claims that it can model human reasoning and therefore be learnable by simple ML models. But following that, it is claimed that because of the success of ML models, it is apparent that human judgement is structured and therefore learnable seems to be circular reasoning because there is no formal measure of human judgement and it is difficult to model.
* The main claim rests on relativization but the paper fails to explain the relativization and the connection to it.
* The verifier, ie the human, is highly efficient/more efficient than the prover which could be seen as a limitation of the theory or a restriction. (Could also remove this)
* Presentation: It is contextualised fine with respect to prior work but the writing style is far from ideal, there are unclear statements, there are lots of lists where it wasn’t necessary.

---

> ### Author Rebuttal · Authors · 2026-03-31
>
> Thank you for your comments! We would like to respond to some of the points that you raised.
>
> > No experimentation or validation of the runtime efficiency guarantees
>
> We agree that experimental validation is an important next step. However, we view our contribution as conceptual. We wanted to send a message to the ML community that debate, which previous works have relied on, is actually not necessary, and to present single-prover interactive proofs as an alternative. As such, our contribution is to define a framework of single-prover interactive proofs for scalable oversight and to prove its feasibility via theoretical results. Our hope is that our work will serve as a theoretical foundation for future works that run experiments.
>
> > The main claim rests on relativization but the paper fails to explain the relativization and the connection to it
>
> A relativizing interactive proof is an interactive proof for oracle-aided computation. Namely, computations such that each gate is either a “basic” AND, OR, NOT gate, or an oracle gate. This is indeed the computation model that this paper focuses on. The notion of oracle-aided computation is defined formally in the preliminaries, and informally in the introduction. We define our model of doubly-efficient interactive proofs for oracle-aided computation in Section 3.
>
> > The verifier, ie the human, is highly efficient/more efficient than the prover which could be seen as a limitation of the theory or a restriction. (Could also remove this)
>
> We assume that the verifier is more efficient, i.e., runs in less time, than the prover since otherwise there would be no need for the verifier to engage in an interactive proof with the prover; the verifier could just perform the computation by himself (without interacting with the prover). So in fact interactive proofs are only interesting to study in this setting where the verifier is more efficient.
>
> > Are there plans to incorporate this with foundational models or agentic systems?
>
>
> Yes, this would be a natural direction for future work. Our work is intended for the training phase, so we could view the prover as a foundational model that is trained to produce an output together with a label for the output, and the verifier as a smaller model that engages in an interactive protocol (that might rely on/make calls to human judgment) with the prover to verify that this label is correct.
>
> > Is using this for reinforcement learning one of the goals?
>
> Yes. In reinforcement learning we need to reward the actions of the agent, and our protocol gives a way to do this efficiently. The idea is now, instead of computing the reward directly, we could have the agent propose a reward, and then we could engage in an interactive protocol with the agent to verify that the reward is accurate. This would be much more efficient than computing the reward ourselves.
>
> > Assuming the prover is honest
>
> We want to clarify that we don’t assume the prover is honest. Our protocols are secure even if the prover is dishonest (if the prover is dishonest, the verifier will reject). This is captured by the “soundness” guarantee of our protocols.

---

> > ### Author Rebuttal · Reviewer_WvV4 · 2026-04-03
> >
> > Thank you for your response. Unfortunately, my concerns still remain. I understand you submitted this paper as a conceptual effort, but I don't see a meaningful contribution in that direction.

---

### Official Review · Reviewer_NBam · 2026-03-13

**Soundness:** 4
**Presentation:** 2
**Significance:** 3
**Originality:** 3
**Overall Recommendation:** 5
**Confidence:** 2

**Summary:**

This work extends the line of work on doubly-efficient interactive proofs from Irving et al. by investigating the question of if we can do away with two model debate, instead using a single-prover. The paper answers the general question in the negative, but shows efficient verification is still achievable in the two limited settings: robust computation and with oracles that are low-degree polynomials.

**Compliance With Llm Reviewing Policy:**

Affirmed.

**Key Questions For Authors:**

N/A

**Limitations:**

yes

**Strengths And Weaknesses:**

1) In terms of significance, the work fits in nicely with previous work, and answers a fairly interesting question within the line of work.

2) The presentation of the work could use some improvement. It assumes a significant amount of background knowledge from the previous line of work. Appendix B helps with this but it is still sometimes difficult to parse. A page of the main paper is spent on trivial preliminaries, defining Language etc, while leaving much to appendix B.

3) The work appears technically sound, with proofs for all major theorems provided in the appendix.

---

> ### Author Rebuttal · Authors · 2026-03-31
>
> Thank you for your comments!

---

> > ### Author Rebuttal · Reviewer_NBam · 2026-04-04
> >
> > N/A

---

### Decision · Program_Chairs · 2026-04-30

**Decision:**

Accept (regular)

**Comment:**

Reviewers agreed that this paper tackles an important topic in AI safety: verification of AI models outputs using interactive proofs. The paper contributes a new theoretical understanding of interactive proofs for AI safety, and new valuable protocols. Although some reviewers are concerned about the applicability to realistic AI safety tasks of the proposed framework, others see this as an important thought-provoking development which can help advance the theory and practice of AI models. Also the area chair support this view and recommend acceptance.


A further comment: the authors should consult the work of Aaronson and Wigderson on Algebrization, which is relevant to their low-degree oracle setting